EMBO
reports

# TPGS1 regulates central spindle microtubule glutamylation and remodeling during telophase and abscission

Rachel Sachs, Yusuke Ogi & Rytis Prekeris [ID] ✉

## Abstract

Microtubules perform a variety of cellular functions, including regulation of mitotic cell division, cilia formation, and neurite extension. Post-translational modifications controlled by the TTLL-family of enzymes confer a host of properties that affect microtubule dynamics and function. Specifically, polyglutamylation of tubulin C-terminal tails plays an important role in regulating microtubule dynamics and function within specific cellular contexts. In this study, we examined contributions from and potential regulators of polyglutamylation during mitosis, focusing on the microtubule remodeling that occurs in telophase once the mitotic spindle has completed chromosome separation. We demonstrate that the anaphase-to-telophase transition is accompanied by an increase in short-chain polyglutamylation of central spindle microtubules. We also show that TTLL1 and TPGS1, subunits of the tubulin polyglutamylation complex, are targeted to the intracellular bridge and midbody during cell progression through telophase. Finally, we demonstrate that loss of TPGS1 leads to defects in remodeling of the central spindle during telophase and impacts the cell's ability to complete mitotic cell division.

**Keywords** Microtubules; Glutamylation; Remodeling; Telophase; Cytokinesis
**Subject Categories** Cell Cycle; Post-translational Modifications & Proteolysis

## Introduction

Cell division is foundational to the development and propagation of life and involves several coordinated and complex processes happening at the same time. During mitosis, mitotic spindle microtubules serve to scaffold and pull apart chromosomes to create the foundation of the two new nuclei. Once chromosomal separation is completed, mitotic spindle microtubules are severed and extensively remodeled (Connell et al, 2009; Matsuo et al, 2013; D'Avino and Capalbo, 2016). Upon initiation of telophase, cells

form the cytokinetic furrow that starts the physical separation of both daughter cells. The ingression of this cytokinetic furrow leads to the formation of the intercellular bridge (ICB) that contains the remnants of the central spindle microtubules from the mitotic spindle (Schiel and Prekeris, 2010; D'Avino and Capalbo, 2016). These highly organized anti-parallel microtubule bundles form the midbody (MB) and ICB, structures that play key roles in orchestrating abscission, the final event in mitotic cell division (Schiel and Prekeris, 2010; Farmer and Prekeris, 2022). While we are beginning to understand the molecular machinery governing microtubule dynamics and function during formation of the mitotic spindle, what regulates microtubule remodeling during the anaphase-to-telophase transition and abscission remains to be fully understood and is the focus of this study.

Many studies have demonstrated that post-translational modifications of free C-terminal tails on both α- and β-tubulin play an important role in regulating microtubule dynamics and function (Janke and Magiera, 2020; Bodakuntla et al, 2021; Chen and Roll-Mecak, 2023). These post-translational modifications are often referred to as the tubulin code and are known to regulate protein association with tubulin, in turn affecting the assembly, disassembly, stability, and conformation of microtubules (Janke and Magiera, 2020; Roll-Mecak, 2020). In this study, we focus on one type of modification, known as glutamylation. Glutamylation is the addition of glutamate onto typically glutamic acid residues within the tubulin C-terminal tail (van Dijk et al, 2007). Due to the negative charge, the addition of glutamate alters the electrostatic interactions of modified tubulin with other proteins (Edde et al, 1990). Glutamylation exists as either monoglutamylation (a single glutamate modification), or polyglutamylation (chains of glutamates added onto one another). Recent studies have suggested that, depending on the length of a polyglutamylation chain, different effects on microtubules can be observed (Valenstein and Roll-Mecak, 2016). Long-chain polyglutamylation is associated with stable microtubules, and is found on mitotic spindles, axonal microtubules, and in cilia (Janke and Magiera, 2020; Roll-Mecak, 2020; Chen and Roll-Mecak, 2023). Long-chain polyglutamylation can help to facilitate motor protein transport, and the modified microtubules are resistant to spastin and/or katanin-dependent severing (Sirajuddin et al, 2014; Valenstein and Roll-Mecak, 2016; Lessard et al, 2019). In contrast, short-chain polyglutamylation has been suggested to induce microtubule severing presumably by

Department of Cell and Developmental Biology, University of Colorado Anschutz Medical Campus, Aurora, CO, USA. ✉E-mail: rytis.prekeris@ucdenver.edu

facilitating microtubule binding to spastin (Lacroix et al, 2010; Valenstein and Roll-Mecak, 2016).

Microtubule glutamylation is regulated by two families of proteins. The tubulin tyrosine ligase-like (TTLL) family contains several glutamylases that catalyze the addition of glutamate chains to tubulin and other proteins (Janke et al, 2005). These chains are removed by proteins in the cytosolic C-peptidase (CCP) family (Rogowski et al, 2010; Chen and Roll-Mecak, 2023). In previous work from our lab, we performed RNA-seq on post-abscission midbodies (MBsomes or midbody remnants) and detected a wide variety of mRNAs that are enriched and translated at the midbody (Farmer and Vaeth et al, 2023). Within that dataset, we found that the mRNA for tubulin polyglutamylation subunit 1 (TPGS1) was enriched in MBsomes. TPGS1 has been previously characterized as the localizing subunit of the tubulin polyglutamylation complex (TPGC), the enzymatic activity of which is facilitated by the glutamylase TTLL1 (Regnard et al, 2003; van Dijk et al, 2007). The presence of TPGS1 mRNA in MBsomes prompted us to look further into the role of glutamylation in telophase and cytokinetic abscission.

In this study, we examined tubulin glutamylation during the anaphase-to-telophase transition, as well as abscission. We observed that during telophase, central spindle microtubules are enriched in short-chain polyglutamylation. We also demonstrated that TTLL1 and its binding protein TPGS1 both localize to central spindle microtubules and the MB. Finally, we demonstrate that loss of TPGS1 leads to central spindle microtubule remodeling defects that prolong mitosis and delay abscission. Taken together, our study indicates a potential role of glutamylation and the TTLL1/TPGS1 complex in mediating microtubule remodeling during the anaphase-to-telophase transition and abscission.

# Results

## Central spindle microtubules undergo remodeling during anaphase-to-telophase transition and midbody formation

Cell progression from metaphase through telophase and ultimately to abscission involves extensive and dynamic remodeling of microtubules (Schiel et al, 2011). This includes packing, cross-linking, and stabilization of central spindle microtubules during the formation of the intercellular bridge (ICB). Central spindle microtubules are arranged to form anti-parallel microtubule bundles that eventually give rise to the midbody (MB) (Fig. 1A) (Schiel et al, 2011). To better understand this process, we used anti-α-tubulin and anti-acetylated α-tubulin antibodies to visualize the rearrangement of central spindle microtubules during the anaphase-to-telophase transition in HeLa cells.

First, we decided to establish the criteria that will be used to determine the stage of telophase progression (early or late telophase) in fixed HeLa cells. It is now well established that cells round up as they enter metaphase. The cells remain round in early telophase (immediately after completion of ingression and formation of the midbody) until they progress to late telophase, when they start flattening out just before entering abscission (Appendix Fig. S1A,C). The second criterion we used is the shape and size of the nucleus. While cells reform the nucleus at the end of anaphase, the nucleus is reniform until cells enter late telophase, when it

circularizes and becomes less compact (Appendix Fig. S1B,D,H). Thus, we used nucleus circularity and size as a second criterion to determine whether a cell is in early or late telophase.

In early telophase, microtubule bundles outside of the ICB are still spread apart (fan-out), presumably because they are in the process of being compacted/remodeled into anti-parallel microtubule bundles (Fig. 1A–D; Appendix Fig. S1F,G). By late telophase, central spindle microtubules have been compacted and are now present almost exclusively within the ICB (Fig. 1B). This process is reflected by a decrease in the ratio of fan-out width to ICB length (ICB aspect ratio), along with the ratio of the minus-end microtubule fan-out to plus-end microtubule widths moving closer to 1 (ICB width ratio) (Fig. 1C,D).

It was suggested that the formation of the anti-parallel microtubule bundles also results in stabilization of microtubules within the ICB. In addition, the ICB microtubules are also heavily acetylated, especially at late telophase (Figs. 1B and 2E) (Janke and Magiera, 2020), another indication of increased microtubule stability (Portran et al, 2017). How these extensive rearrangements of central spindle microtubules are regulated remains to be fully understood and is the focus of this study.

## Cell progression to telophase leads to a decrease in long-chain polyglutamylation of central spindle microtubules

C-terminal post-translational modification of α- and β-tubulins has been implicated in regulating microtubule function and dynamics. Polyglutamylation is one of these modifications and is thought to be directly involved in microtubule dynamics and potentially spastin-mediated microtubule severing during abscission (Lacroix et al, 2010; Valenstein and Roll-Mecak, 2016). Thus, next we decided to examine ICB and mitotic spindle microtubule polyglutamylation during mitosis. Since the extent of C-terminal modification can vary from a single glutamate to long and sometimes branched glutamate chains, we also decided to investigate what type of glutamylation occurs on ICB microtubules. To that end, we used two different types of anti-glutamylation antibodies. Monoclonal recombinant tubulin anti-glutamylation antibody rGT335 is known to detect tubulin glutamylation of 1 or more glutamates encompassing most glutamylated chains, while anti-polyglutamylation antibody PolyE detects polyglutamylation with 4 or more glutamate chains of any protein, including tubulin (Wolff et al, 1992; Lacroix et al, 2010; Blasius et al, 2025).

Consistent with previous studies (Janke and Magiera, 2020), acetylation of ICB microtubules increased as cells progressed through telophase (Fig. 2A,B,E). In contrast, long-chain polyglutamylation (visualized with the PolyE antibody) is only detected during metaphase and disappears by the time cells progress to telophase (Fig. 2B,D; Appendix Fig. S3). To our surprise, the rGT335 signal remained constant in metaphase, early telophase, and late telophase microtubules (Fig. 2A,C; Appendix Fig. S2A–F). These data suggest that while the overall extent of tubulin glutamylation does not dramatically change in telophase, it does appear to switch to short-chain (3 or less glutamates) glutamylation, at least within the ICB microtubules.

Our data suggest that progression from metaphase to telophase involves a switch to short-chain glutamylation. What remains unclear is what molecular mechanisms drive this change in glutamylation. One possibility is that CCPs may shorten glutamylation chains by removing

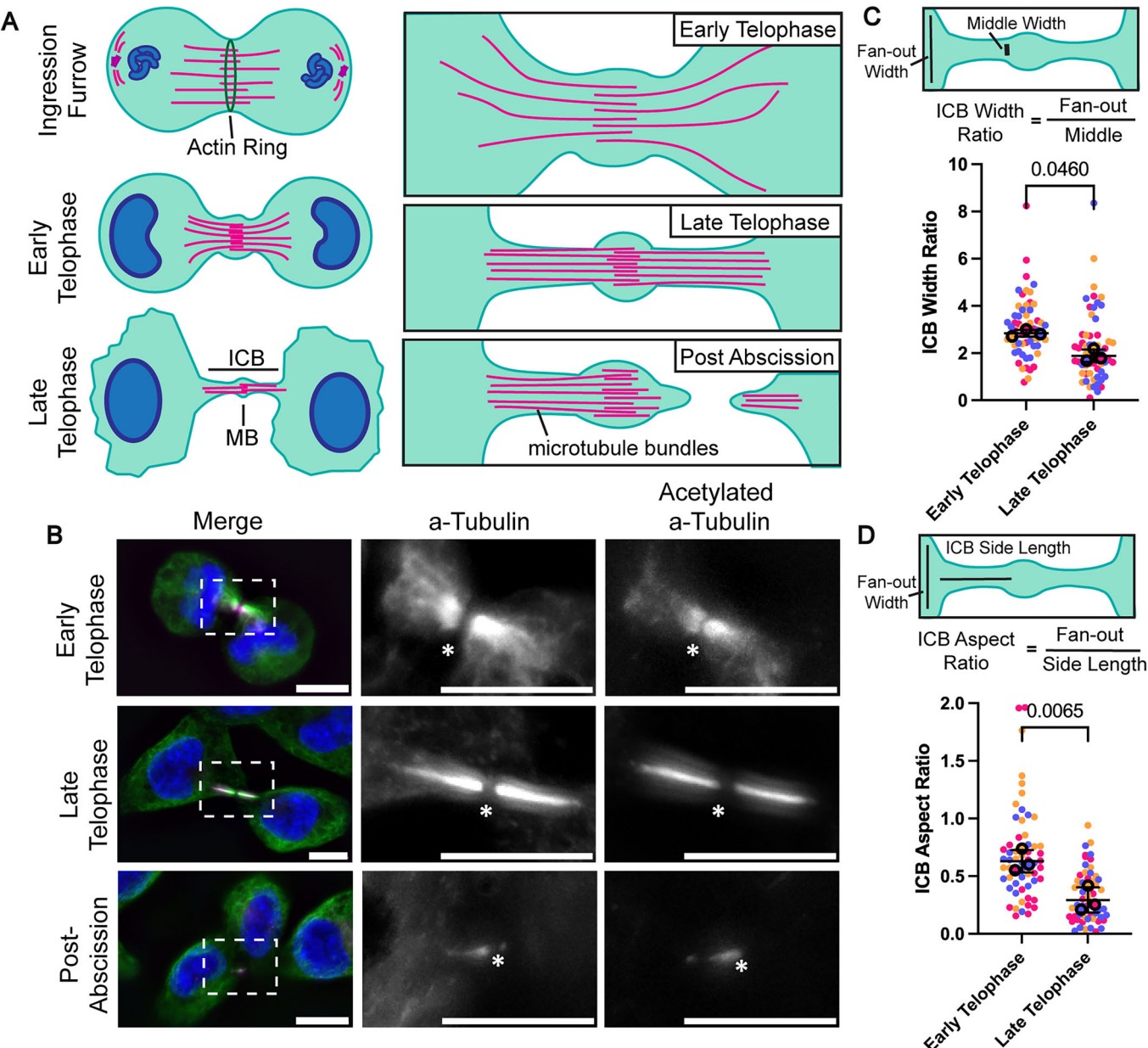

**Figure 1. Central spindle microtubules undergo remodeling during anaphase-to-telophase transition and midbody formation.**

(A) Model of central spindle microtubule remodeling during cytokinesis and abscission. After separation from the centrosome, anti-parallel microtubules remain in the midzone, where the actin-myosin ring forms the ingression furrow. As cells progress to early telophase, these microtubules begin to be reorganized into anti-parallel microtubule bundles. By late telophase, the now ICB microtubules are bundled and crosslinked together. (B) HeLa cells were fixed and stained with anti-α-tubulin (green) and anti-acetylated α-tubulin (magenta) antibodies. The box in the left image indicates areas that are zoomed-in on the right. All scale bars are 10 μm. An asterisk marks the MB. (C, D) Quantification based on images from (B). Measurements were done on the left or top-most side of the ICB. Ten cells for each mitotic stage were analyzed for each biological replicate (three replicates, color-coded). Statistics were calculated with Student's *t* test on median values for each stage. Median was used where technical replicates do not meet standard distribution testing in place of means, calculated using normality and log normality tests. Error bars represent the SEM surrounding median values. Source data are available online for this figure.

some of the glutamates, thus shortening glutamate chain length during telophase. Another option is that microtubules are completely de-glutamylated during anaphase, and then new shorter glutamate chains are added as cells enter telophase. A third possibility is that while most of the mitotic spindle microtubules are glutamylated, the non-kinetochore central spindle microtubules that become the ICB microtubules are not. In this case, the short-chain glutamylation is then added to ICB microtubules sometime between anaphase and early telophase. To test these possibilities, we next examined tubulin glutamylation during early stages of furrow ingression by measuring colocalization between anti-acetylated α-tubulin and recombinant GT335 antibody staining in central spindle microtubules (Fig. 2F,G).

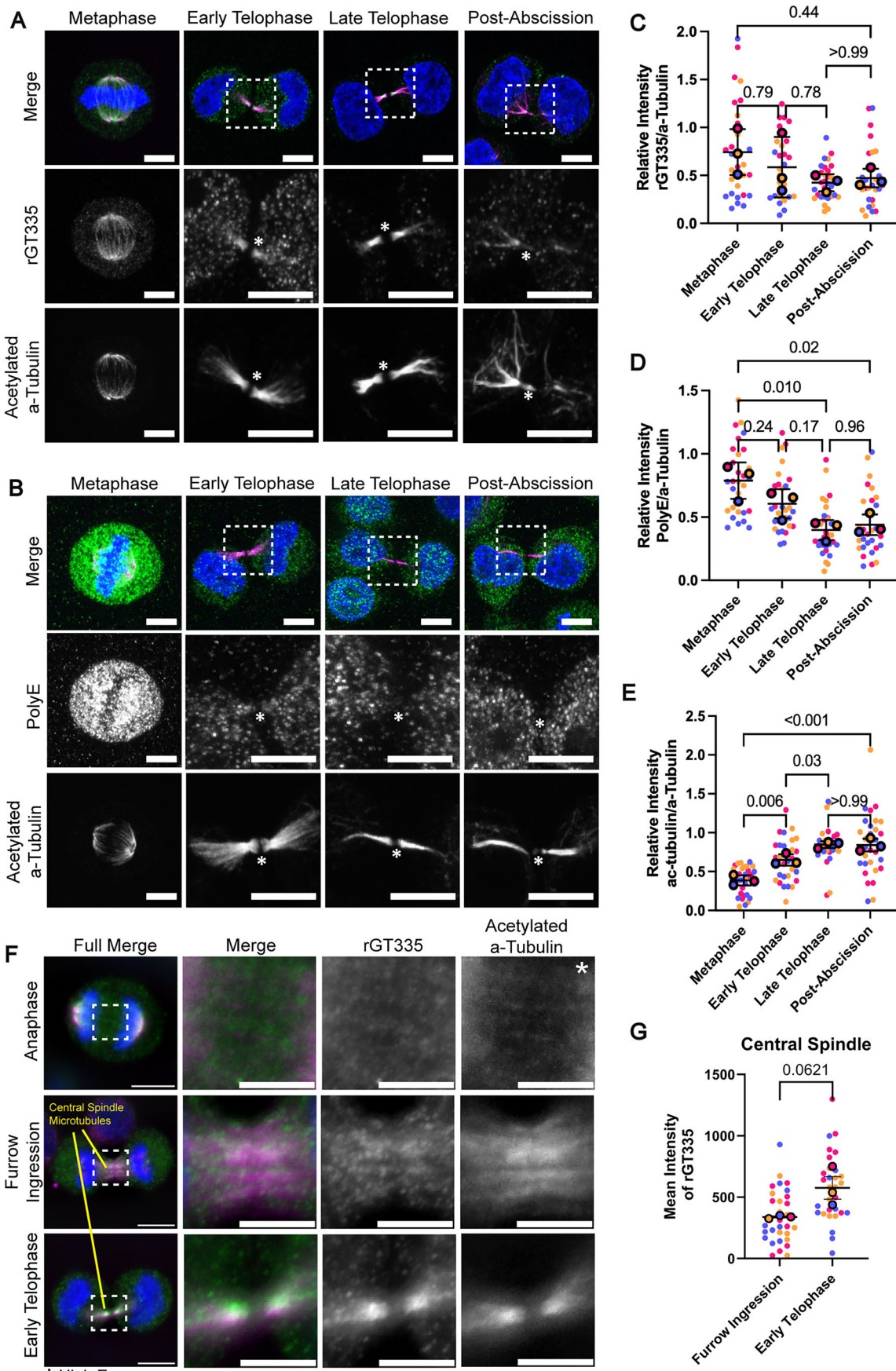

Figure 2. Cell progression to telophase leads to a decrease in long-chain polyglutamylation of central spindle microtubules.

(A, B) HeLa cells were fixed and stained with anti-acetylated α-tubulin (magenta) and recombinant GT335 (green) antibodies (A) or anti-acetylated α-tubulin (magenta) and PolyE (green) antibodies (B), and confocal images were taken. Boxes indicate zoomed-in sections. All scale bars are 10 μm. An asterisk marks the MB. (C–E) HeLa cells were fixed and co-stained with anti-α-tubulin and either rGT335 (C), PolyE (D), or anti-acetylated α-tubulin (E) antibodies. Mean fluorescence intensity of antibodies was measured from microtubule structures and then normalized to the corresponding α-tubulin signal. Each biological replicate ($n = 3$) measures ten cells per stage, coded by color. One-way ANOVA was calculated on the means of each set of replicates along with a Dunnett post-test. Error bars represent SEM of biological replicates. (F) HeLa cells were fixed and stained with anti-acetylated α-tubulin (magenta) and recombinant GT335 (green) antibodies. Boxes indicate zoomed-in sections. Scale bars are 10 μm on full images, and 5 μm on zoomed-in images. Asterisk marks an image of acetylated α-tubulin in anaphase, where a higher brightness was used to better visualize microtubules in the region of interest. (G) Quantification of furrow ingression and early telophase cells from (F). Cells were co-stained with rGT335 and anti-acetylated α-tubulin antibodies. Mean fluorescence intensity of antibodies was measured from central spindle microtubule structures identified through acetylated α-tubulin staining and taken from the corresponding rGT335 region. Each biological replicate ($n = 3$) measures ten cells per stage, coded by color. Student's $t$ test was calculated on the means of each set of replicates. Error bars represent SEM of biological replicates. Source data are available online for this figure.

Interestingly, the rGT335 signal seems to be much lower on central spindle microtubules (as compared to kinetochore microtubules) and gradually increased during furrow ingression and early telophase (Fig. 2F,G). Importantly, a strong rGT335 signal can still be observed on kinetochore microtubules during anaphase (Fig. 2F), suggesting that glutamylation of microtubules in the central spindle is regulated differentially from kinetochore microtubules. These data suggest that central spindle microtubules are likely not glutamylated during anaphase and that short-chain glutamylation is added to central spindle tubulin as cells progress to telophase.

## TTLL1 is enriched at the central spindle microtubules in late telophase

Our data suggest that short-chain glutamylation is added on central spindle microtubules as cells progress towards telophase and abscission. Tubulin polyglutamylation is regulated by two different protein families. Glutamate can be added to the C-terminus of α- and β-tubulin by several members of the TTLLs (Fig. 3A) (Janke et al, 2005). In contrast, tubulin C-terminal glutamates can be removed by CCPs (Chen and Roll-Mecak, 2023). Interestingly, TTLL glutamylases prefer modifying microtubules (Garnham et al, 2015), while CCPs appear to predominantly catalyze glutamate removal from α/β-tubulin heterodimers (Chen and Roll-Mecak, 2023). To determine which TTLLs and CCPs may function to mediate changes in ICB tubulin glutamylation, we next asked which CCPs and TTLLs are present within the midbody and ICB. Since CCP1 and CCP5 appear to be the main erasers of tubulin polyglutamylation present in HeLa cells, we transfected cells with EYFP-CCP1 or CCP5-EYFP and analyzed their localization during telophase. As shown in Appendix Fig. S4D–F, neither EYFP-CCP1 nor CCP5-EYFP could be detected within the ICB, suggesting that CCP1 and CCP5 are unlikely to play a role in regulating ICB microtubule reorganization by telophase. Consistent with that, while overexpression of CCP1 and CCP5 did reduce tubulin glutamylation (Appendix Fig. S4A–C), it did not appear to have any effect on remodeling ICB microtubules during telophase progression (Appendix Fig. S4D–F).

Next, we analyzed the localization of all TTLL glutamylases during cell division (Fig. 3A). To that end, we performed a transient overexpression screen of EGFP or EYFP-tagged TTLLs and found that TTLL1 and TTLL11 were both enriched at the midbody during telophase (Fig. 3B,C). Interestingly, TTLL1-EGFP seems to be most enriched by late telophase, as determined by staging telophase cells based on nucleus shape (Fig. 3D,E). If we stage cells based on cell shape (flattened cells representing late telophase while round cells representing early telophase) we also find that 78.1% of flattened

telophase cells (presumably in late telophase) had TTLL1 at the MB. In contrast, only 35.7% of round telophase cells (presumably in early telophase) had TTLL1 at the MB. All these data suggest that TTLL1 may play a role in mediating ICB microtubule rearrangement during telophase progression. Thus, for the rest of this study, we will focus on TTLL1 and its effect on microtubule remodeling during the anaphase-to-telophase transition.

Next, we wanted to confirm TTLL1 localization during telophase by testing whether endogenous TTLL1 is also present in the MB. To that end, we isolated post-abscission MBs (known as MBsomes or MB remnants) from HeLa cells and tested for the presence of TTLL1 in the MBs by Western blotting. As shown in Fig. 4A, endogenous TTLL1 is present in MBsomes, suggesting that TTLL1 could be involved in central spindle microtubule modification during telophase and abscission.

## TPGS1 localizes to central spindle microtubules during telophase

Prior work in our laboratory discovered that a specific population of mRNAs, as well as translation machinery, are all present in MBs (Farmer and Vaeth et al, 2023). This study also indicated that certain abscission-mediating proteins can be locally translated at the MB during cytokinetic abscission, and that this local MB-associated translation is required for protein targeting to the MB. To determine whether TTLLs may also be translated at the MB, we examined the MB-associated RNA-seq dataset for the presence of TTLL and CCP mRNAs. We found that none of the mRNAs encoding TTLLs and CCPs were enriched at the MB, suggesting that local MB-associated translation is not directly involved in targeting TTLL1 to the MB (Table 1). There was however one exception, that being the mRNA encoding for TPGS1, which was enriched threefold in relative mRNA abundance compared to the whole-cell transcriptome (Table 1).

TPGS1 has originally been identified as part of the tubulin polyglutamylation complex (TPGC) that includes TTLL1 (van Dijk et al, 2007). Unlike other TTLL family members, TTLL1 does not directly bind to microtubules and requires the TPGC complex for its enzymatic activity. Importantly, TPGS1 has been previously shown to serve as the localizing subunit in the complex with TTLL1 (Regnard et al, 2003; van Dijk et al, 2007) (Fig. 4B). Given that TTLL1-EGFP was enriched in the MB, we hypothesized that the localized translation of TPGS1 mRNA may function to target TTLL1 to the ICB during telophase.

To explore the possibility that TPGS1 may mediate TTLL1 targeting to the ICB/MB, we next decided to investigate the

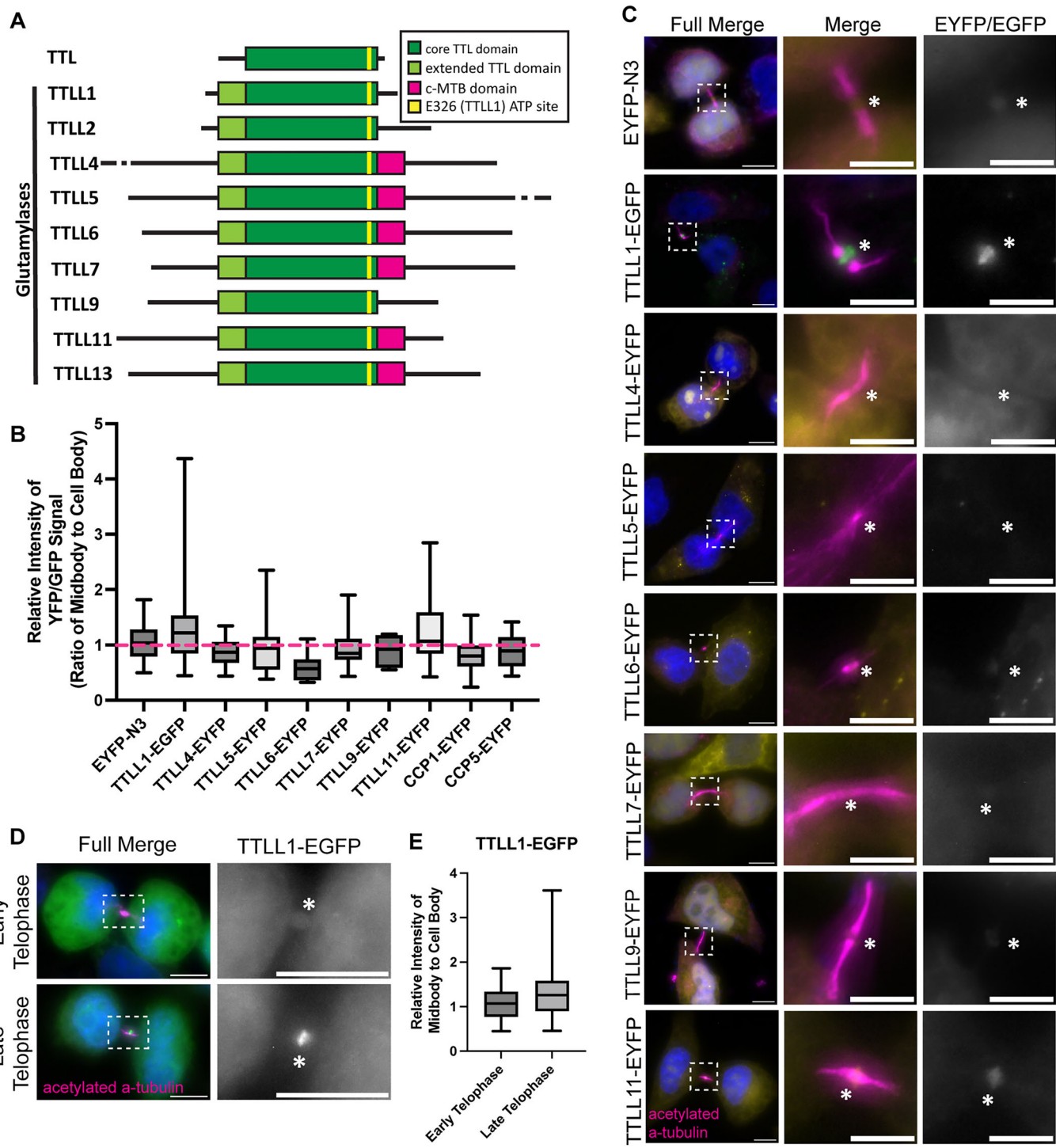

**Figure 3. TTLL1 is enriched at the central spindle microtubules in late telophase.**

(A) Schematic representation of human glutamylases belonging to the TTLL family. (B, C) Analysis of localization during telophase. HeLa cells were transiently transfected with an EYFP or EGFP-tagged TTLLs or CCPs, fixed, and stained with anti-acetylated α-tubulin antibodies (magenta) to identify cells in telophase. Each box plot (min-max) represents a single biological replicate, with data points representing technical replicates of a minimum size of $n = 5$ as could be obtained from the biological replicate. The box plots show the minimum and maximum values, with the box showing the mean and 25%/75% quartiles. The enrichment of TTLLs and CCPs at the MB was calculated by measuring the ratio of the mean intensity of EYFP/EGFP signal in the MB to the cell body (minus the nucleus). Boxes indicate zoomed-in sections. Full image scale bars are 10 μm, zoomed-in scale bars are 5 μm. An asterisk marks the MB. Statistics were not calculated between groups. (D) The localization of TTLL1-EGFP in early and late telophase cells. Boxes indicate zoomed-in sections. All image scale bars are 10 μm. An asterisk marks the MB. (E) TTLL1-EGFP samples sorted based on stage from the screen. Each box plot represents the min-max data of the technical replicates from one biological replicate, same as in (B). Statistics were not calculated between groups. Source data are available online for this figure.

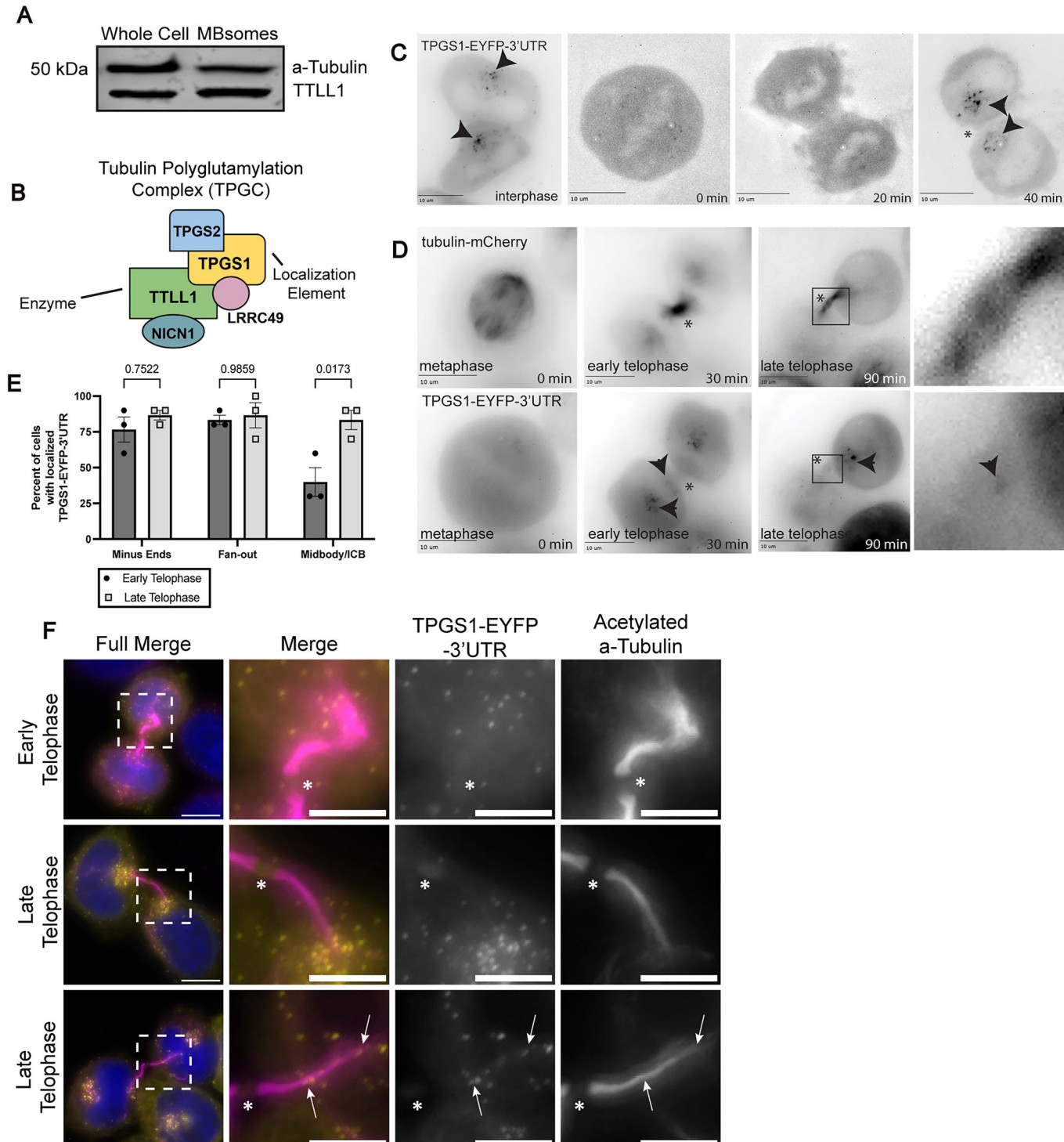

localization of TPGS1 during mitotic cell division. The enrichment of TPGS1 mRNA at the MB suggests that TPGS1 may be locally translated at the MB and our prior work has shown that the 3'UTR is often required for correct localization of proteins encoded by MB-enriched mRNA (Farmer and Vaeth et al, 2023) (Table 1). Thus, we generated a TPGS1-EYFP construct containing the TPGS1 3'UTR (TPGS1-EYFP-3'UTR) and analyzed its localization

during telophase. As shown in Fig. 4F, TPGS1-EYFP-3'UTR could be observed in the ICB and the MB during late telophase.

To better understand the spatiotemporal dynamics of TPGS1 during mitotic cell division, we next used time-lapse microscopy to visualize TPGS1-EYFP-3'UTR in live dividing HeLa cells. Interestingly, in interphase cells TPGS1-EYFP-3'UTR was mostly present in small puncta that accumulated around centrosomes (Fig. 4C, see

**Figure 4.  TPGS1 is required for targeting TTLL1 to the midbody.**

(A) Western blot analysis of whole-cell lysates and isolated MBsomes, both loaded with 10 μg of protein as measured by Bradford assay. Blot was probed sequentially with anti-TTLL1 and anti-α-tubulin antibodies, respectively. (B) Diagram of Tubulin Polyglutamylation Complex (TPGC) members as described in Regnard et al (2003). (C) Time-lapse imaging of an interphase HeLa cell progressing into mitosis, shown left to right as interphase, metaphase, furrow ingression, and early telophase, and transfected with TPGS1-EYFP-3'UTR. Arrows mark defined TPGS1-EYFP-3'UTR puncti, and all scale bars are 10 μm. (D) Time-lapse imaging of a metaphase HeLa cell progressing to late telophase. From left to right, metaphase, early telophase, late telophase, and a zoomed in image of late telophase are shown. Cell is co-transfected with tubulin-mCherry (top) and TPGS1-EYFP-3'UTR (bottom). Arrows indicate localization of TPGS1-EYFP-3'UTR puncti at ICB minus ends in panel 3, and MB localization in panel 4. (E, F) Quantification of TPGS1-EYFP-3'UTR in early and late telophase cells, shown in F stained with anti-acetylated α-tubulin (magenta). Cells in each category were counted based on the presence or absence of puncta at that location per image (10 cells per biological replicate, three biological replicates) and expressed as a percent of images per replicate. Two-way ANOVA was performed and shown statistics represent the p-value from early to late telophase per location. Error bars represent the SEM. In (F), full image scale bars are 10 μm, zoomed-in scale bars are 5 μm. Asterisk marks the MB, and arrows point to puncta localized within the ICB. Source data are available online for this figure.

arrows). These puncta mostly disappeared as cells entered metaphase and only reformed once cells entered early telophase (Fig. 4C,D). Consistent with our fixed cell imaging data (Fig. 4E,F), we could also detect TPGS1 puncta in the ICB and MB (Fig. 4D). Curiously, many TPGS1-EYFP-3'UTR puncta could also be observed at the minus-ends of the ICB microtubules (Fig. 4C–F). These minus-ends of ICB-microtubules are part of fan-out ICB edges observed in early telophase (Fig. 1), suggesting that TPGS1 may be involved in mediating the remodeling of these fan-out microtubules into anti-parallel microtubule bundles as cells progress through telophase.

While both TTLL1 and TPGS1 could be observed associating with ICB microtubules, their distribution was somewhat different. TTLL1 is mostly observed at the MB, while TPGS1 in addition appears to associate with fan-out parts of ICB microtubules, although some of TPGS1 can also be observed within the ICB and MB. One possibility is that minus-end-associated TPGS1 may complex with other TTLLs, such as closely related TTLL9. Alternatively, since TPGS1 was suggested to be a localizing subunit of TTLL1/TPGS1 complex, overexpression of TTLL1 alone does not fully recapitulate the localization of endogenous (and TPGS1-bound) TTLL1. Unfortunately, cells did not tolerate well the co-overexpression of both TTLL1 and TPGS1. It is worth noting that generally, cells also tolerated only low overexpression of TTLL1 or TPGS1, suggesting that TTLL1 is likely a very low-abundance enzyme in most cells. Consequently, we could not differentiate between those two possibilities, and further work will be needed to fully understand the molecular machinery governing TTLL1 and TPGS1 localization.

## TPGS1 is required for targeting TTLL1 to the midbody

To directly test whether TPGS1 is needed for TTLL1 localization to the MB during telophase, we generated a TPGS1 knock-out (TPGS1-KO) HeLa cell line (Appendix Fig. S5). Next, we tested the effect of TPGS1 KO on the localization of TTLL1-EGFP during mitotic cell division. Since cells poorly tolerate overexpression of active wild-type TTLL1, we generated a TTLL1-EGFP-E326G mutant containing a glutamic acid-to-glycine substitution at the TTLL1 ATP-active site to limit off-target effects (Wang et al, 2022). Importantly, TTLL1-E326G-EGFP was also enriched in MB during late telophase (Fig. 5A,B), suggesting that TTLL1-E326G localization mimics the localization of wild-type TTLL1. As shown in Fig. 5A,B, TTLL1-E326G-EGFP was no longer targeted to the MB in TPGS1-KO cells, indicating that TPGS1 is required for TTLL1 localization at the ICB during telophase.

## TPGS1 is required for central spindle microtubule remodeling during telophase

Our data have shown that TPGS1 localizes within the ICB, as well as at the minus-ends of ICB microtubules. Thus, we wondered whether TPGS1 may affect cell division. To that end, we performed time-lapse analysis of control and TPGS1-KO cells. As shown in Fig. 5C, TPGS1-KO cells appear to divide slower as compared to control cells. Interestingly, TPGS1 KO did not affect the time required for the metaphase-to-anaphase transition (Fig. 5D,E), suggesting that TPGS1 is likely required during telophase.

To further investigate what effect TPGS1 KO has on ICB microtubule remodeling during telophase progression, we next analyzed microtubule dynamics by transfecting cells with GFP-tubulin and visualizing cell division by time-lapse microscopy. As previously reported, in control cells during early telophase microtubules were rapidly compacted into anti-parallel microtubule bundles that accumulated in the newly formed ICB (Fig. 5F, top row) (Schiel and Prekeris, 2010). In contrast, in TPGS1-KO cells, ICB microtubules remained fanned out much longer than in control cells (Fig. 5E, bottom row).

To further validate this observation, we fixed control and TPGS1-KO cells and visualized endogenous microtubules using anti-α-tubulin and anti-acetylated α-tubulin antibodies. Consistent with our time-lapse data, TPGS1-KO cells exhibited increased ICB microtubule fan-out in late telophase (Fig. 6A–G; Appendix Fig. S6A–D). Interestingly, due to ICB microtubules remaining uncompacted, the minus-ends of the ICB microtubule bundles could sometimes be observed to split to accommodate the position of the nucleus (Fig. 6A,D). While this is rarely observed in control cells, over 30% of TPGS1-KO cells exhibit this phenotype (Fig. 6D). Importantly, overexpression of TPGS1-EYFP-3'UTR partially rescued the defect in ICB microtubule organization in TPGS1-KO cells (Fig. 6E–G). Taken together, our data indicate that TPGS1 and likely TTLL1 are involved in mediating microtubule remodeling during the anaphase-to-telophase transition.

## TPGS1-KO cells have reduced tubulin glutamylation in the ICB and exhibit abscission defects

Since it was proposed that TPGS1 facilitates TTLL1 localization and therefore activity at specific sites, we hypothesized that we may see some change in microtubule glutamylation in TPGS1-KO cells. To investigate this, we stained fixed control and TPGS1-KO cells with anti-α-tubulin and rGT335 antibodies. Consistent with the

Table 1. Results of the RNA-seq experiments performed in Farmer and Vaeth et al (2023) for TTLL glutamylases, CCPs, known complex members, and severing proteins.

| Target | Mean Abundance | | T-Test | Enrichment Trend | Standard Deviation | |
|---|---|---|---|---|---|---|
| | Whole Cell | Midbody | | | Whole Cell | Midbody |
| TPGS1 | 13.545 | 39.834 | 0.010 | Enriched | 5.085 | 8.637 |
| TTLL4 | 36.696 | 3.003 | 0.011 | De-enriched | 12.318 | 4.294 |
| TTLL5 | 28.066 | 8.396 | 0.048 | De-enriched | 5.590 | 10.741 |
| TTLL6* | 0.364 | 0.000 | 0.007 | De-enriched | 0.122 | 0.000 |
| TTLL7 | 6.983 | 0.317 | 0.001 | De-enriched | 1.010 | 0.549 |
| CCP1 | 7.371 | 2.197 | 0.032 | De-enriched | 1.916 | 1.996 |
| Spastin | 4.615 | 0.820 | 0.024 | De-enriched | 1.539 | 1.060 |
| KATNB1 | 30.221 | 9.032 | 0.031 | De-enriched | 3.762 | 10.659 |
| TTLL1* | 1.238 | 3.510 | 0.557 | N.S. | 0.889 | 6.079 |
| TTLL11 | 4.480 | 8.697 | 0.390 | N.S. | 0.383 | 7.565 |
| TPGS2 | 120.194 | 105.999 | 0.675 | N.S. | 25.752 | 48.035 |
| LRRC49 | 5.109 | 6.086 | 0.862 | N.S. | 0.518 | 9.136 |
| NICN1* | 1.389 | 1.843 | 0.818 | N.S. | 0.167 | 3.192 |
| CSTPP1 | 50.161 | 43.692 | 0.739 | N.S. | 14.928 | 27.641 |
| CCP5 | 29.216 | 12.568 | 0.103 | N.S. | 12.411 | 5.844 |
| KATNA1 | 15.519 | 22.556 | 0.722 | N.S. | 0.870 | 31.859 |
| *Samples whole cell mRNA measurements fall below threshold of 50 counts but are non-zero | | | | | | |

Mean abundance controls for the difference in total mRNA contents between whole cells and MBsomes and is representative of biological replicates ($n = 3$). $T$ test was performed on abundance scores. Standard deviation is given to show replicate variability. N.S. indicates no significance. Green marks enriched mRNA, red indicates mRNA less present in the midbody than the whole cell, and blue shows no difference in the relative mRNA detected between the midbody and whole cell.

involvement of TPGS1/TTLL1 in glutamylation of central spindle microtubules, in both early and late telophase, TPGS1-KO cells showed a marked reduction in ICB microtubule glutamylation (Fig. 7A,C,D; Appendix Fig. S6E,F). Importantly, TPGS1 KO did not lead to a decrease in rGT335 signal during metaphase, suggesting the role of TPGS1/TTLL1 during cell division is largely restricted to telophase (Fig. 7B,E; Appendix Fig. S6G,H).

Our data indicate that TPGS1 mediates ICB microtubule remodeling, thus directly affecting the cell's ability to complete abscission and mitotic division. Next, we wanted to determine which aspect of abscission is affected by TPGS1-KO. There are three major possibilities of how defects in ICB microtubule remodeling may affect abscission. First, depletion of TPGS1 may affect the cross-linking of the microtubules, thus delaying the progression of cells to late telophase. Second, defects in ICB microtubule organization may affect organelle transport to the MB. Third, defects in microtubule organization and glutamylation may affect the recruitment and/or enzymatic activity of spastin, the microtubule severing enzyme that is required for abscission.

To differentiate between all these possibilities, we first tested whether endocytic trafficking to the MB was perturbed, as glutamylation is known to affect transport along microtubules (Sirajuddin et al, 2014). To test this, we transfected cells with GFP-FIP3, a known marker for MB-associated endosomes (Schiel et al, 2012) and analyzed the localization of GFP-FIP3-containing endosomes during telophase both by time-lapse imaging (Appendix Fig. S7A) and in fixed cells staining with anti-acetylated α-tubulin (Appendix Fig. S7B,C). It was previously demonstrated that FIP3-endosomes localize to centrosomes during metaphase and anaphase (Schiel et al, 2012). As cells progress to the telophase, FIP3-endosomes translocate to the MB by moving along ICB microtubules, where they regulate actin depolymerization at the abscission site (Schiel et al, 2012). These dynamics were not affected in TPGS1-KO cells (Appendix Fig. S7B), indicating that TPGS1-mediated ICB microtubule glutamylation does not appear to affect microtubule-based transport, although we cannot fully rule out the possibility that the speed and efficiency of FIP3-endosome transport were affected in TPGS1-KO cells.

Next, we examined whether TPGS1-KO affects cross-linking of ICB microtubules during the progression to telophase. To that end, we stained fixed control and TPGS1-KO cells with anti-α-tubulin and anti-PRC1 antibodies. PRC1 is a known microtubule cross-linker involved in microtubule compaction during telophase (Subramanian et al, 2010; Gaska et al, 2020). As shown in Appendix Fig. S8, PRC1 intensity within the ICB remained unchanged, indicating that TPGS1/TTLL1-dependent microtubule glutamylation does not affect PRC1 localization to microtubules during mitotic cell division.

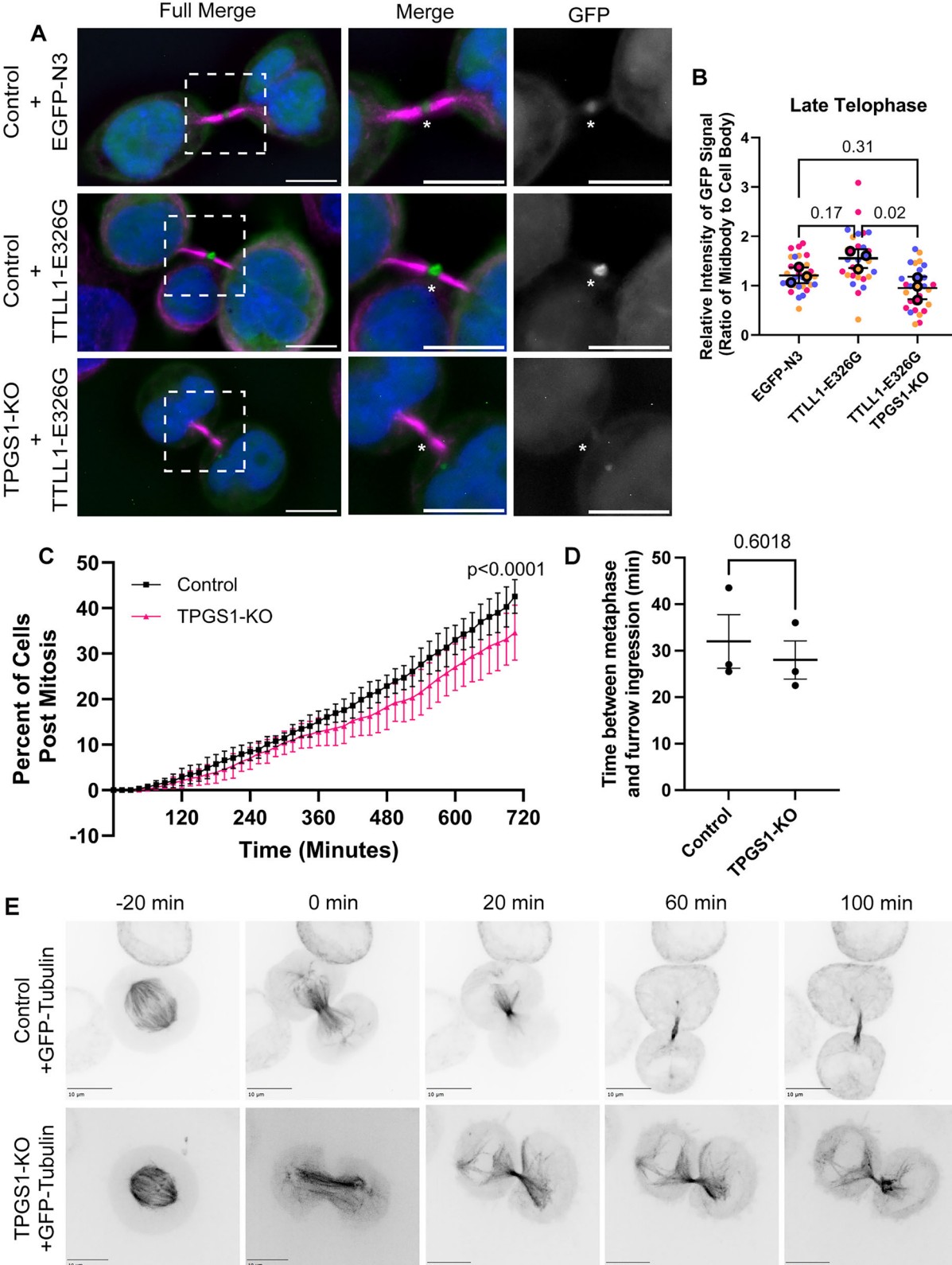

**Figure 5. TPGS1 and TTLL1 co-localize to central spindle microtubules and MB during telophase.**

(A, B) Quantification of images (B) of control and TPGS1-KO HeLa cells transfected with either EGFP-N3 or TTLL1-E326G-EGFP (green), fixed and stained with anti-acetylated α-tubulin (magenta). The enrichment of EGFP signal at the MB was calculated by measuring the ratio of the mean intensity of EGFP signal in the MB to the cell body (minus the nucleus). Each biological replicate ($n = 3$) measures ten cells per stage, coded by color. One-way ANOVA was calculated on the means of each set of replicates along with a Dunnett post-test. Error bars represent SEM of biological replicates. For representative images in (A), boxes indicate zoomed-in sections. All image scale bars are 10 μm. An asterisk marks the MB. (C) Live-cell brightfield imaging was used to follow control and TPGS1-KO cells over 12 h, with images taken in z-stacks (step size 1 μm) every 15 min. At each time point, the percentage of cells in each of 5 points on the coverslip that had finished dividing was counted, with each field treated as one biological replicate. Data was analyzed using non-linear regression analysis with error bars representing the SEM between the 5 set points. (D) From live-cell imaging in (C), ten cells (technical replicates) from different quadrants in three fields of view were timed from T0 (metaphase) to the start of furrow ingression. Each point represents the average of each technical replicate, and each field of view was treated as one biological replicate. Statistics were done with Student's $t$ test between control and TPGS1-KO cells. (E) Time-lapse imaging of GFP-tubulin transfected control and TPGS1-KO cells, at intervals of 20 min. T0 represents the start of furrow ingression and cells were followed for 100 min, with representative images shown at T-20 (metaphase), T0 (furrow ingression), T20 (approximately early telophase for both), T60 and T100 (approximately late telophase for control cells). All image scale bars are 10 μm. Source data are available online for this figure.

Lastly, we fixed cells and used anti-acetylated α-tubulin and anti-spastin antibodies to analyze spastin localization and function during abscission. We evaluated the localization of spastin to the ICB and found that spastin localization does not appear to be changed in TPGS1-KO cells (Appendix Fig. S9). Katanin is another microtubule-severing enzyme that is known to mediate mitotic spindle remodeling during division, although katanin was reported to predominantly function at the centrosome rather than the ICB (McNally et al, 2006). Consistent with these reports, we did not observe katanin at the ICB during telophase, and its localization at centrosomes and minus-ends of ICB microtubules was largely unaffected by TPGS1-KO (Appendix Fig. S10). However, further and more extensive analysis will be needed to fully rule out the effect of TPGS1-KO on spastin and/or katanin localization during telophase.

To examine whether spastin-dependent severing during abscission is disrupted in TPGS1-KO cells, we next determined the fraction of late telophase cells with a clearly identifiable abscission cut in ICB microtubules (Fig. 7F). As shown in Fig. 7G, TPGS1 KO led to a decrease in the number of cells completing microtubule severing. All these data suggest that TPGS1/TTLL1-dependent glutamylation of ICB microtubules likely plays an important role in regulating spastin-dependent microtubule severing during abscission (Fig. 8).

## Discussion

The mitotic spindle plays a key role in ensuring the fidelity of chromosome separation during mitotic cell division. Mitotic spindle function depends on highly controlled and locally regulated changes in microtubule dynamics, a process that has been extensively studied during the last couple of decades. Importantly, upon completion of anaphase, the remnants of the mitotic spindle, which consists predominantly of central spindle microtubules, also undergo extensive remodeling (Fig. 1A). This remodeling includes severing central spindle microtubules and compacting/reorganizing these microtubules into highly organized anti-parallel microtubule bundles that form ICB microtubules and MBs. Despite the importance of ICB microtubules and MBs in regulating cleavage furrow ingression and abscission (Schiel and Prekeris, 2010; Farmer and Prekeris, 2022), the molecular machinery governing central spindle microtubule remodeling during the anaphase-to-telophase transition remains poorly understood. Similarly, we still know little about how the selection of the abscission site within ICB microtubules is regulated. Understanding these processes was one of the goals of this study.

Post-translational tubulin modifications have emerged as a key regulator of microtubule dynamics and function and are sometimes referred to as the tubulin code (Janke and Magiera, 2020; Roll-Mecak, 2020). Here, we show that one of these modifications, glutamylation, appears to play an important role during central spindle remodeling and abscission. Interestingly, the glutamate chain length appears to affect the function of tubulin glutamylation. For example, short-chain glutamylation was proposed to mediate spastin-dependent microtubule severing (Lacroix et al, 2010; Valenstein and Roll-Mecak, 2016). Meanwhile, long-chain poly-glutamylated microtubules correlate with increased microtubule stability through resistance to spastin severing (Chen and Roll-Mecak, 2023). Furthermore, long-chain polyglutamylation was shown to regulate dynein and kinesin-based transport (Sirajuddin et al, 2014; Lessard et al, 2019). Our data suggest that ICB and MB microtubules appear to be predominantly modified via short-chain polyglutamylation. Consistent with that, we show that a decrease in ICB microtubule polyglutamylation leads to defects in spastin-dependent abscission. Interestingly, spastin recruitment to the abscission site did not appear to be affected by the decrease in ICB microtubule glutamylation. This result is in line with prior work on ESCRT-III and spastin localization to the abscission site (Yang et al, 2008; Connell et al, 2009), indicating that spastin localization is not dependent on short-chain polyglutamylation, but its activity might be.

Interestingly, TTLL1 and TPGS1 are typically associated with α-tubulin glutamylation (Janke and Magiera, 2020; van Dijk et al, 2007). However, some studies suggested that β-tubulin glutamylation is the main modulator of spastin severing (Valenstein and Roll-Mecak, 2016). Similarly, though longer-chain polyglutamylation was suggested to activate spastin-dependent microtubule severing (Valenstein and Roll-Mecak, 2016), our results indicate that central spindle microtubules during telophase are predominantly modified with short-chain (< 4 glutamates) polyglutamylation. One possibility explaining these differences is that aforementioned studies have been performed in vitro and it remains to be demonstrated whether these observations also apply in cello and in vivo. Future research will be needed to help expand our understanding of how severing is regulated by glutamylation state.

In our work, we also aimed to understand the dynamics underlying a switch to short-chain polyglutamylation during the anaphase-to-telophase transition. Previous studies have shown that glutamylation is predominantly added onto polymerized microtubules and removed from free tubulin heterodimers (Chen and

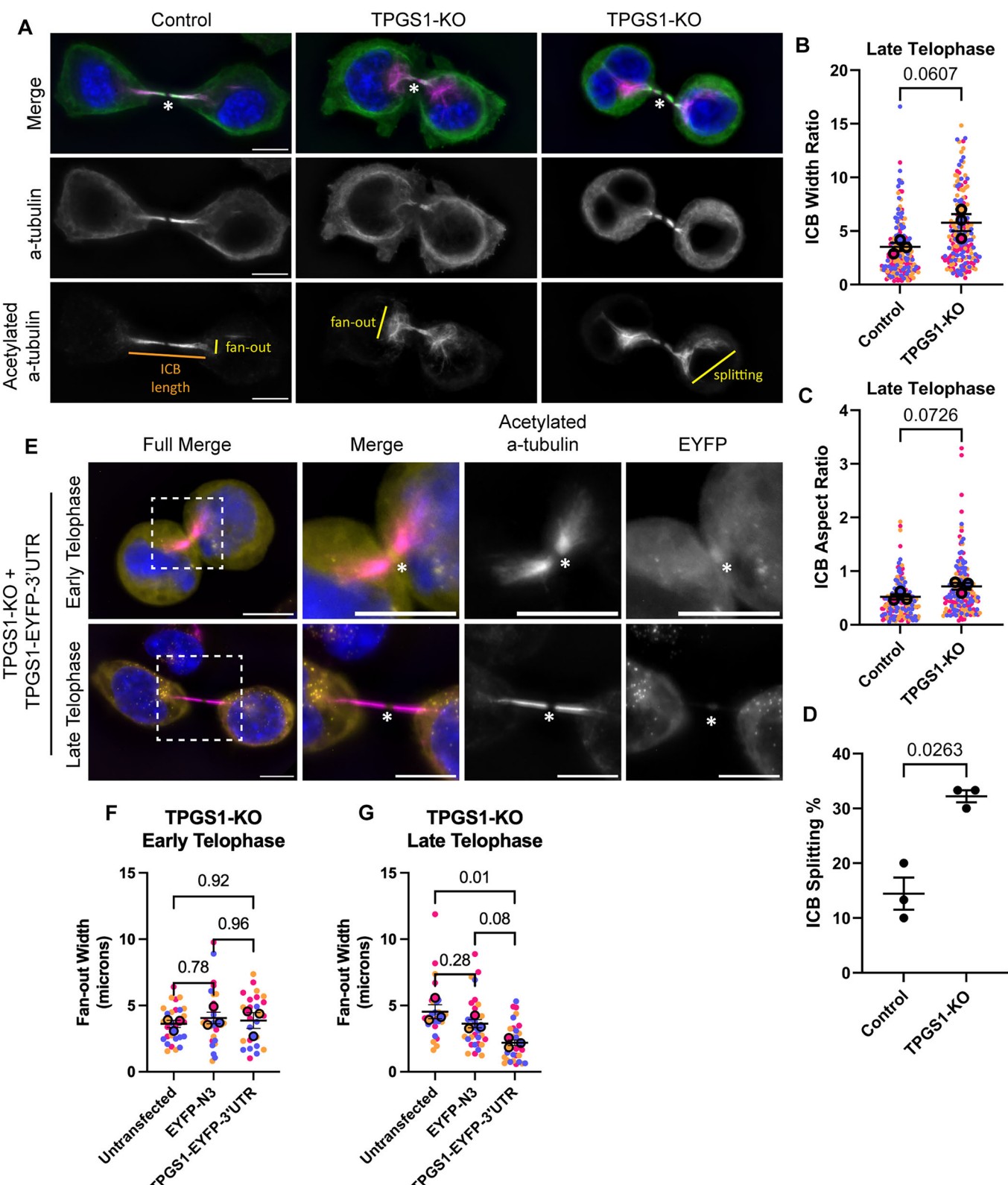

**A** Control | TPGS1-KO | TPGS1-KO
Merge / a-tubulin / Acetylated a-tubulin (fan-out, ICB length, splitting labels)

**B** Late Telophase — ICB Width Ratio — 0.0607 — Control vs TPGS1-KO

**C** Late Telophase — ICB Aspect Ratio — 0.0726 — Control vs TPGS1-KO

**D** ICB Splitting % — 0.0263 — Control vs TPGS1-KO

**E** TPGS1-KO + TPGS1-EYFP-3'UTR — Full Merge | Merge | Acetylated a-tubulin | EYFP — Early Telophase / Late Telophase

**F** TPGS1-KO Early Telophase — Fan-out Width (microns) — 0.92, 0.78, 0.96 — Untransfected, EYFP-N3, TPGS1-EYFP-3'UTR

**G** TPGS1-KO Late Telophase — Fan-out Width (microns) — 0.01, 0.28, 0.08 — Untransfected, EYFP-N3, TPGS1-EYFP-3'UTR

Roll-Mecak, 2023). ICB microtubules are compacted into anti-parallel microtubule bundles that are suggested to be very stable (Schiel and Prekeris, 2010; Matsuo et al, 2013; D'Avino and Capalbo, 2016). That limits the extent of polymerization/ depolymerization cycles within ICB microtubules during telophase, consequently limiting CCP-dependent removal of polyglutamylation. Thus, we hypothesized that any change in glutamylation state of central spindle microtubules is likely to occur during anaphase

◀ **Figure 6. TPGS1 is required for central spindle microtubule remodeling during telophase.**

(A) Representative images of late telophase control and TPGS1-KO HeLa cells. Cells were fixed and stained with anti-α-tubulin (green) and anti-acetylated α-tubulin (magenta) antibodies. All image scale bars are 10 μm. An asterisk marks the MB. (B, C) Quantification of images from (A). Measurements were done on each side of the ICB from the MB for each cell. Thirty cells for each mitotic stage were analyzed for each biological replicate (three biological replicates, color-coded), with each side of the midbody as a technical replicate. Statistics are calculated with *t* test on median values for each stage. Median is used where technical replicates do not meet standard distribution testing in place of the mean. Error bars represent the SEM surrounding median values. (D) Quantification of images from (A). In each of three biological replicates, the percentage of control and TPGS1-KO late telophase cells that split the fan-out minus ends around the nucleus was calculated (30 cells per replicate). *T* test was calculated on the values from each set of biological replicates. Error bars represent SEM. (E) Representative images of early and late telophase TPGS1-KO cells transiently transfected with TPGS1-EYFP-3'UTR (yellow) and stained with anti-acetylated α-tubulin (magenta). All image scale bars are 10 μm. Asterisk marks the MB, and boxes indicate the zoomed-in area. (F, G) Quantification of images from (E). Up to ten cells each per stage of three biological replicates were quantified in each condition of TPGS1-KO cells: transfected with EYFP-N3, transfected with TPGS1-EYFP-3'UTR, and untransfected from the TPGS1-EYFP-3'UTR condition. Fan-out width was measured on the top or left-most side of the ICB. One-way ANOVA with a Tukey post-test was conducted on the three groups. Source data are available online for this figure.

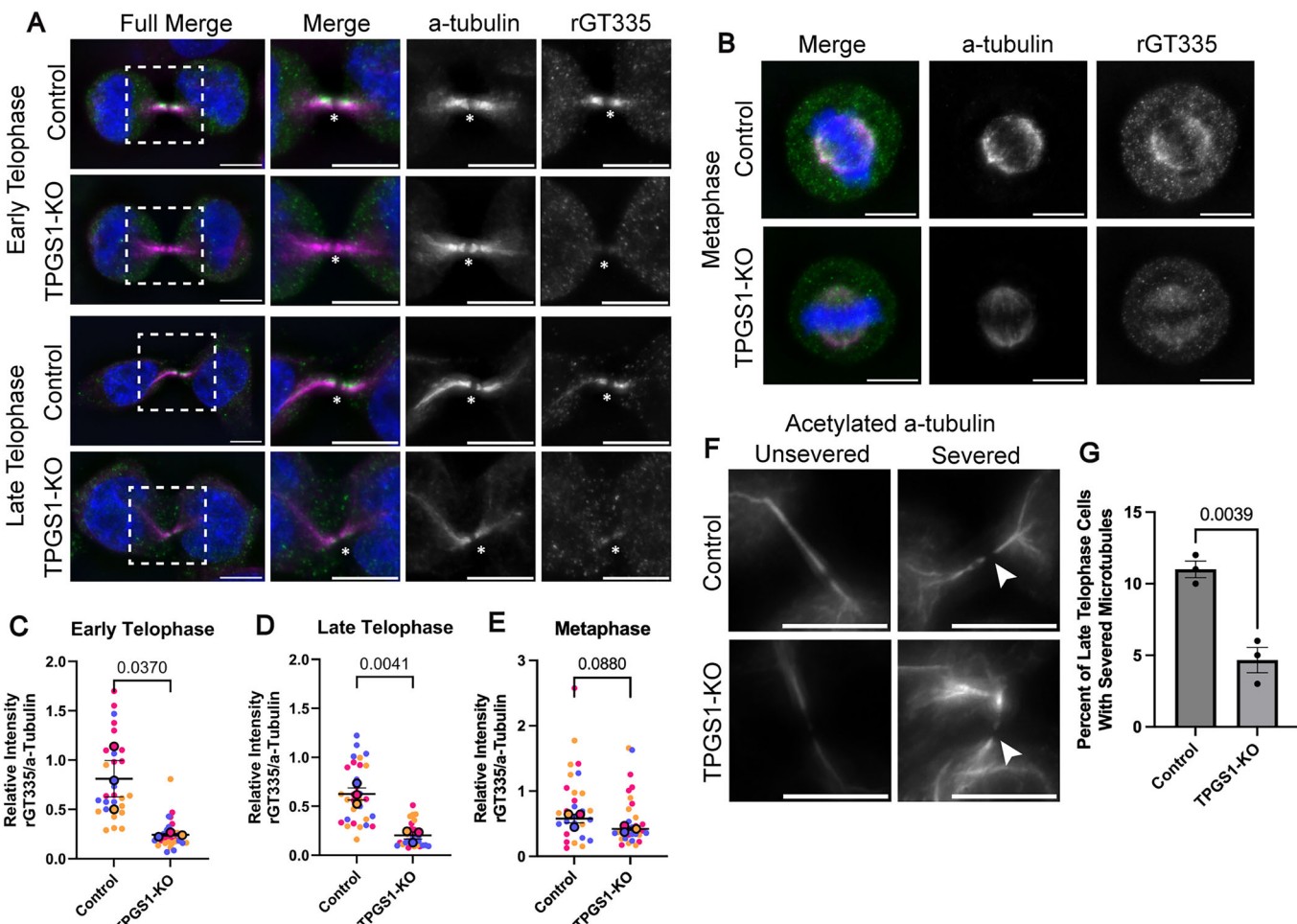

**Figure 7. TPGS1-KO cells have reduced tubulin glutamylation in the ICB and exhibit abscission defects.**

(A–E) Imaging and quantification of early and late telophase (A, C, D), and metaphase (B, E) control and TPGS1-KO HeLa cells. Cells were fixed and co-stained with anti-rGT335 (green) and anti-α-tubulin (magenta) antibodies. All image scale bars are 10 μm. Asterisk marks the MB, and boxes represent the zoomed-in area. Biological replicate ($n = 3$) means are color-coded, with 10 cells per replicate. Statistics are calculated with *t* test on mean values for each stage. Error bars represent the SEM surrounding mean values. (F) Fixed wild-type HeLa cells, indicating unsevered and severed microtubules in pre-abscission late telophase ICBs for control and TPGS1-KO cells. Cells are stained with an anti-acetylated α-tubulin (gray) antibody. Image scale bars are 10 μm. Arrows mark the microtubule severing site. (G) Quantification of severing sites in pre-abscission late telophase control and TPGS1-KO HeLa cells. *T* test was done on the percent of cells (100 cells counted per replicate) with a severing site between three biological replicates. Error bars represent the SEM. Source data are available online for this figure.

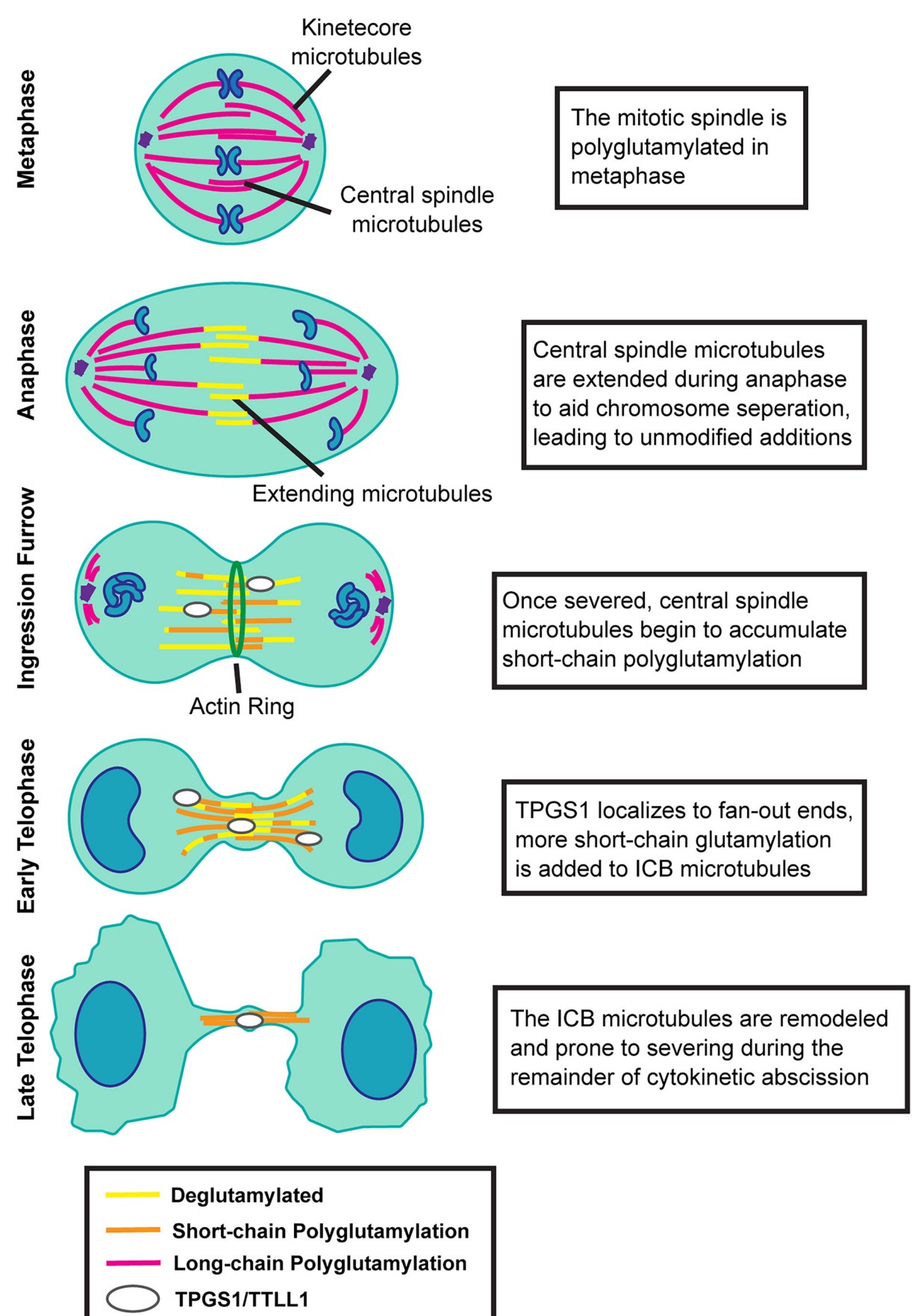

**Figure 8.  A model of TPGS1-mediated microtubule remodeling and glutamylation in mitosis.**

Schematic representation of changes in microtubule glutamylation state from anaphase through late telophase.

and before these microtubules are compacted into stable ICB microtubule bundles. Consistent with this hypothesis, we found that central spindle microtubules have accumulated short-chain polyglutamylation during furrow ingression and continue to accumulate it into early telophase. Thus, although further research is needed to understand the differences between metaphase and anaphase spindle polyglutamylation, we suggest that polyglutamylation is not present on central spindle tubulin during metaphase and is added to elongating microtubules, potentially starting in anaphase and continuing through furrow ingression and telophase. These events presumably lead to enrichment of ICB microtubules in short-chain polyglutamylation. It is of interest then that TTLL1 (and TTLL11, which was also enriched at the midbody in our initial screen), is classified as primarily an elongator (Janke et al, 2008). Thus, it remains to be determined whether TTLL1 can also function as an initiator during telophase microtubule remodeling or whether another TTLL glutamylase is responsible for mediating short-chain glutamylation of central spindle microtubules.

If ICB microtubules are polyglutamylated during telophase, the question then is what mediates this modification. In our screen of tubulin code writers (TTLLs) and erasers (CCPs), we found that TTLL1-EGFP was enriched within the ICB and more specifically at the MB. This result was of interest considering the enrichment of TPGS1 mRNA seen in MBsomes. In our previous work, we demonstrated that local translation of MB-associated mRNAs is required for abscission (Farmer and Vaeth et al, 2023). Thus, we hypothesize that TTLL1 enrichment at the ICB microtubules may be due to localized translation of TPGS1, since TPGS1 is known to be a targeting subunit of the TTLL1-containing TPGC (Regnard et al, 2003; van Dijk et al, 2007), and that MB and ICB localization of TTLL1 may facilitate short-chain glutamylation of microtubules during telophase and abscission. It is of note to us that the MB enrichment of TPGS1 that we found previously was not shared by a similar study, Park et al (2023). While we are unsure exactly what caused the discrepancy, our prior study and Park et al (2023) did use different methods of MB isolation. In Farmer and Vaeth et al (2023), post-mitotic MBsomes were isolated for RNA-sequencing, whereas Park et al (2023) isolated midbodies pre-abscission. This timing difference may account for the unaligned observations, as it's possible TPGS1 did not yet accumulate in MB in their study.

Interestingly, there seems to be some difference in the localization of TTLL1 and TPGS1 in mitotic cells. TTLL1-EGFP is present almost exclusively in the MB with relatively little present in other parts of the ICB. In contrast, while TPGS1 is also present at the MB, the majority of TPGS1 puncta are associated with the minus ends (fan-out) of ICB microtubules, especially during early stages of microtubule remodeling into anti-parallel microtubule bundles. At least some of these differences in localization could be related to the overexpression of individual TPGC subunits. We, however, cannot rule out the possibility that TPGS1 puncta associated with minus ends of microtubules may be part of a TPGC that does not have TTLL1. Indeed, TTLL9 is closely related to TTLL1, and it is possible that TPGS1 may form distinct complexes with either TTLL1 or TTLL9. Another possibility is that

TPGS1-containing targeting complex forms first and then recruits TTLL1 during late telophase. Further work, however, will be needed to test this hypothesis.

The localization of TPGS1 at the fan-out minus ends of ICB microtubules raises an interesting possibility that TPGS1-dependent polyglutamylation may also regulate central spindle microtubule remodeling during the anaphase-to-telophase transition. Consistent with this hypothesis, TPGS1-KO cells not only exhibit delays in abscission but also defects in the formation of anti-parallel ICB microtubule bundles. How TPGS1-dependent polyglutamylation affects microtubule bundling remains to be fully understood since the association of cross-linking proteins, such as PRC1, appears to be unaffected. Remodeling of microtubules during telophase is attributed to activities of severing enzymes like spastin and katanin (Connell et al, 2009; Matsuo et al, 2013; D'Avino and Capalbo, 2016). As a decrease in polyglutamylation disrupts microtubule severing at the abscission site, a similar mechanism may lead to the lack of microtubule severing and pruning by spastin or katanin during the anaphase-to-telophase transition. Importantly, similar defects in the formation of ICB microtubule bundles were observed in spastin-depleted HeLa cells (Schiel et al, 2011). Whether ICB microtubule bundling is also regulated by TTLL1, or possibly TTLL9, remains unknown since we were never able to generate TTLL1-KO or efficient (>75%) TTLL1 knockdown cells.

In this study, we highlight a cellular role for short-chain glutamylation-dependent microtubule remodeling during the anaphase-to-telophase transition, as well as microtubule severing during abscission. Additionally, we show that TPGS1 is required for both of these processes, where it likely functions to recruit and activate the TTLL1 glutamylase. Identification of the involvement of glutamylation in regulating ICB microtubules lays a foundation for further studies in the function of the tubulin code during mitosis.

## Methods

### Reagents and tools table

| Reagent/resource | Reference or source | Identifier or catalog number |
|---|---|---|
| **Experimental models** | | |
| HeLa (*H. sapiens*) | ATCC | |
| **Recombinant DNA** | | |
| pTTLL1-EGFP | Gift from Dr. Carsten Janke | |
| pTTLL4-EYFP | Gift from Dr. Carsten Janke | |
| pTTLL5-EYFP | Gift from Dr. Carsten Janke | |
| pTTLL6-EYFP | Gift from Dr. Carsten Janke | |

| Reagent/resource | Reference or source | Identifier or catalog number |
|---|---|---|
| pTTLL7-EYFP | Gift from Dr. Carsten Janke | |
| pTTLL11-EYFP | Gift from Dr. Carsten Janke | |
| pCCP1-EYFP | Gift from Dr. Carsten Janke | |
| pCCP5-EYFP | Gift from Dr. Carsten Janke | |
| pEYFP-N3 | Clontech | |
| pEGFP-N3 | Clontech | |
| pTTLL1-E326G-EGFP | This study | |
| pTPGS1-EYFP-3'UTR | This study | |
| **Antibodies** | | |
| Mouse anti-α-tubulin (B512) | Invitrogen | Cat # 32-2500, IF 1:100 WB 1:1000 |
| Rabbit anti-α-tubulin | Abcam | Cat # AB15246, 1:100 |
| Rabbit anti-Acetylated Tubulin (K40) | Cell Signaling | Cat #5335, 1:200 |
| Mouse anti-Acetylated Tubulin | Sigma-Aldrich | Cat #T7451, 1:500 |
| Mouse recombinant anti-GT335 | Gift from Dr. Kristen Verhey | Cite, 1:100 |
| Rabbit anti-PolyE (IN105) | AdipoGen | Cat # AG-25B-0030, 1:100 |
| Rabbit anti-PRC1 | Protein Tech | Cat # 15617-1-AP, 1:100 |
| Mouse anti-Spastin (1:100) (Santa Cruz, sc-53443) | Santa Cruz | Cat # sc-53443, 1:100 |
| Rabbit anti-p60 katanin | Abcam | Cat# AB111881, 1:100 |
| Hoechst 3342 | AnaSpec | Cat # AS-83218, 1:5000 |
| Alexa-488 Anti-Rabbit Secondary | Jackson ImmunoResearch | Cat # 711-545-152, 1:100 |
| Alexa-594 Anti-Mouse Secondary | Jackson ImmunoResearch | Cat #715-585-150, 1:100 |
| Alexa-488 Anti-Mouse Secondary | Jackson ImmunoResearch | Cat # 715-545-150, 1:100 |
| Alexa-594 Anti-Rabbit Secondary | Jackson ImmunoResearch | Cat # 711-585-152, 1:100 |
| IRDye 680RD anti-mouse Secondary | Li-cor | Cat #926-68072, 1:10,000 |
| IRDye 800CW anti-rabbit Secondary | Li-cor | Cat #926-32213, 1:10,000 |
| **Oligonucleotides and other sequence-based reagents** | | |
| TPGS1 qPCR Forward | This study | CGGCGTGACGGAAATGCTA |
| TPGS1 qPCR Reverse | This study | CCATGTTCTCGAAGTAGTGAGC |

| Reagent/resource | Reference or source | Identifier or catalog number |
|---|---|---|
| TTLL1 qPCR Forward | This study | GCTCTCACAGATCAAAAAGTGGT |
| TTLL1 qPCR Reverse | This study | CCGCCAATTAGTAACGGGTTG |
| GAPDH qPCR Forward | This study | CTGGGCTACACTGAGCACC |
| GAPDH qPCR Reverse | This study | AAGTGGTCGTTGAGGGCAATG |
| **Chemicals, enzymes, and other reagents** | | |
| Phusion High-Fidelity DNA Polymerase | New England Biolabs | Cat #M0530S |
| Lipofectamine Transfection Reagent | Invitrogen | Cat # 18324012 |
| tracrRNA | Horizon Discovery | Cat #U-002005-xx |
| DharmaFECT Duo transfection reagent | Horizon Discovery | Cat #T-2010-xx |
| iTaq Universal SYBR Green | Bio-Rad | Cat # 1725120 |
| SuperScript VI Reverse Transcriptase | Invitrogen | Cat # 18090010 |
| The Original TA Cloning Kit | Invitrogen | Cat # K202020 |
| NEBuilder HiFi DNA Assembly | New England Biolabs | Cat # E2621 |
| **Software** | | |
| GraphPad Prism 9 | https://www.graphpad.com/ | |
| FIJI | https://imagej.net/ij/index.html | |
| Adobe Illustrator | Adobe Inc. | |

## Cell culture

HeLa cells used in this study were purchased from ATCC. HeLa cells were cultured at 37 °C with 5% $CO_2$ in DMEM with 1% penicillin/streptomycin and 10% FBS. Cell lines were routinely tested for mycoplasma and authenticated through STR testing.

## Transient over-expression

HeLa cells were transfected with overexpression plasmids using either Lipofectamine or electroporation. Lipofectamine transfections were done according to the manufacturer's instructions. Electroporation was performed using 10 μg of DNA per 10-cm plate. Cells were pelleted and pre-washed in PBS, then resuspended in PBS and put on ice, combined with the plasmid. After 10 min, samples were electroporated at 250 volts, capacity 925, in the

Bio-Rad GenePulser Xcell. Cells were then plated on collagen-coated coverslips and left to recover 24 h before imaging.

## Staging of cells in mitosis and within telophase

Cells were staged in mitosis based on Hoechst nuclear staining and on cell morphology. Early and late telophase cells were identified first by the presence of an intercellular bridge (through either α-tubulin or acetylated α-tubulin signal), then staged based on the nucleus primarily. Early telophase cells were identified by a small reniform nucleus, while late telophase cells had larger and rounder nuclei. Cell shape as seen through α-tubulin staining was used as a secondary indicator. Cells in early telophase were identified by a round cell shape, and late telophase cells were identified by being larger in size and better adhered to the coverslip. Post-abscission is differentiated from late telophase by identification of a free ICB between two cells that is not attached to one of the cells and is absent of pulling tension that otherwise causes a straight structure.

## Generation of CRISPR KO lines

HeLa cells stably expressing Tet-inducible Cas9 were grown in a 6-cm dish to ~75% confluency, followed by treatment with 1 μg/ml doxycycline for 24 h to induce Cas9 expression. Cells were then transfected with crRNA:tracrRNA mix using the DharmaFECT DUO transfection reagent as described by the manufacturer's protocol. Transfected cells were incubated for 24 h, trypsinized, and plated for individual clones. Individual clones were screened by TA cloning using The Original TA Cloning Kit (Invitrogen) and then genotyped by sequencing. In experiments using KO cells, the parental Cas9 line was used as the control.

## Western blot analysis

HeLa cells were lysed on ice in buffer containing 20 mM Tris-HCl, 150 mM NaCl, 1 mM EDTA, 1% Triton X-100, 10% glycerol, and protease inhibitor. After 30 min, lysates were centrifuged at 15,000× g. In total, lysate samples were prepared in 6x SDS loading dye, boiled for 5 min at 95 °C, and 10 μg was loaded onto and separated via SDS-PAGE. Gels were transferred onto a 0.45-μm polyvinylidene difluoride membrane, followed by blocking for 30 min in 5% non-fat milk diluted in tris-buffered saline with 0.1% Tween-20 (TBST). Membranes were then incubated with primary antibodies (diluted in 5% BSA in TBST) overnight at 4 °C or for one hour at room temperature. Membranes were washed 2× in TBST followed by incubation with IRDye-tagged secondary antibodies made in 5% non-fat milk diluted in TBST for 30 min at room temperature. Blots were washed 2X again with TBST before imaging on a Li-Cor Odyssey CLx.

## RNA extraction, cDNA synthesis, and RT-qPCR

RNA extraction was performed using the Trizol reagent according to the manufacturer's instructions. cDNA was generated using SuperScript IV on 1 μg of RNA per sample according to the recommended protocol, using d(T)20 as the synthesis primer. Quantitative PCR was performed using iTaq Universal SYBR Green Supermix together with target-specific primers (TPGS1, GAPDH)

and cDNA. To quantify the RT-qPCR, each cDNA sample was normalized to GAPDH.

## MBsome purification

Post-abscission midbodies were collected from HeLa cell media using the protocol previously described (Peterman and Prekeris, 2017). Briefly, media from HeLa cells were collected and subjected to a series of centrifugation steps (3000×g, 10,000×g). Sucrose gradient fractionation was then performed at 3000×g for 30 min, and the interphase between 40% glycerol and 2 M sucrose was collected. The MBsomes were then sedimented at 10,000×g for 45 min, and the MBsomes were resuspended in 20 μl of PBS for western blot analysis.

## Immunofluorescence microscopy

Cells were plated on collagen-coated coverslips and grown to ~70% confluency. Cells were fixed with 2% paraformaldehyde (PFA) for 15 min at room temperature. PFA was then quenched in phosphate-buffered saline (PBS) with 7.5 mg/ml of glycine. Cells were permeabilized and blocked with PBS buffer containing 4 mg/ml saponin, 0.002 mg/mL BSA, and 2% FBS for 30 min at room temperature. Primary antibodies were diluted in blocking buffer and incubated for one hour at room temperature. Cells were then washed 2× with PBS before adding secondary antibodies for 30 min at room temperature. Cells were washed 2× again, with the first PBS wash containing Hoechst (1:5000). Coverslips were then mounted on glass slides with VectaShield and sealed with nail polish. All images, unless otherwise stated, were acquired on an inverted Zeiss Axiovert 200 M microscope using a ×63 oil objective, QE charge-coupled device camera (Sensicam), and Slidebook v. 6.0 software (Intelligent Imaging Innovations). These images were taken in z-stacks with a step size of 0.5 μm (9 steps). Images in Fig. 2 were acquired using confocal microscopy on the Nikon Eclipse Ti2 inverted A1 confocal microscope with a 63× oil objective and a z-step size of 0.25 μm.

## Time-lapse microscopy

HeLa cells were transfected with either GFP-tubulin or TPGS1-EYFP-3'UTR and plated on collagen-coated 35 mm glass-bottom MatTek dishes. Cells were incubated for 24 h and then imaged using an inverted Zeiss Axiovert 200 M microscope using a ×63 oil objective, QE charge-coupled device camera (Sensicam). To maintain cells at 37 °C during time-lapse imaging, glass-bottom dishes were mounted on Lab-Tek S1 Heating insert. Cells were imaged for a total of 3 h, and 10-plane z-stack (step size 1 μm) was taken at every time point. To limit photodamage, images were taken every 15 min (12 total time points).

## Cell proliferation analysis

Data for Fig. 8C were obtained through live-cell imaging for a period of 12 h with images collected in 15-min intervals, acquired using the Nikon Eclipse Ti2 inverted A1 confocal microscope with a Plan Apo VC 20x DIC N2 objective. HeLa cells were placed in a humidity-controlled chamber maintained at 5% $CO_2$ and 37 °C. Images were taken as an 11-step z-stack with a step size of 1 μm.

## Image analyses

### ICB and cell measurements

Measurements were performed on α-tubulin or acetylated α-tubulin individual channels. Fan-out width was measured perpendicular to the ICB, from the top-most minus-end of the acetylated tubulin signal to the bottom-most tip. The middle width was measured directly on either side of the MB gap in the acetylated tubulin signal, perpendicular to the ICB, from the top of the signal to the bottom. ICB length was measured for the whole bridge through α-tubulin signal, from the middle of the fan-out if present across the ICB over the MB to the middle of the fan-out on the other side, while ICB side length was measured from the middle of the fan-out to the end of the signal at the MB. All measurements were taken from the left or top-most side of the ICB, with the exception of Fig. 6B–D, where both sides were measured. ICB splitting was measured by the presence or absence of microtubules on at least one side curving into two parts around the nucleus, expressed as a percentage of the cells per replicate.

ICB intensity measurements were made through ROI (region of interest) selection of the ICB and other microtubule structures, with mean intensity measurements taken for each channel individually from the same ROI. The background mean intensity of each antibody was subtracted per channel from the obtained mean intensities outside of cells through an ROI box on each channel. In the case of central spindle microtubule measurements, the background was subtracted from within the cell. Results quantifying an antibody of interest were performed on cells co-stained with α-tubulin and normalized to the corresponding α-tubulin channel per image to account for tubulin content, apart from central spindle microtubule measurements in Fig. 2E being taken from only the rGT335 channel. All cells were imaged at the same exposure. In the case of anaphase/furrow ingression measurement that lacks the defined microtubule structures, a round ROI of the midzone of cells was taken where anti-parallel microtubules overlap.

Cell and nucleus area and circularity were measured by selecting the α-tubulin signal and the Hoechst signal to measure total area and perimeter. Circularity was calculated with the formula $\frac{4\pi \times Area}{Perimeter^2}$, with 1 indicating a perfect circle. In the case of telophase cells, the mask was made on the left-most cell in each pair.

Measurements were performed on max projections of z-stacks centered on the ICB with 4um of 0.5um steps on each side. Each of three biological replicates measures ten cells per stage, unless otherwise noted.

### MB enrichment measurements

MB enrichment intensity was measured using an ROI around the MB and cell body (excluding the nucleus). Measurements were calculated in terms of mean intensity for the ROI, with background subtracted as described above. Final calculations were controlled relative to the cell body signal in the same image, calculated by dividing the mean intensity of the MB by the mean intensity of the cell body. Each of three biological replicates measures ten cells per stage, except for the initial screen in Fig. 4B, where only one replicate was performed. Intensity measurements were performed only on the YFP/GFP channel, as antibody staining is unable to penetrate the MB, unlike the signal from tagged proteins.

### Cell division quantification

From live-cell images collected over 12 h, five points of imaging per coverslip were used for quantification, with images taken every 15 min. Total cells (every cell in the image regardless of stage and including interphase) were counted at the first frame per field, and at each time interval the number of cells that had undergone mitosis (visualized as progression over frames from rounding up, going through metaphase/anaphase/furrow ingression, and then flattening back on the coverslip, at which point they were counted as divided) were counted. As each time-point passed, the number of cells divided in that time point was added to the previous time-point and expressed as a percentage of total cells seen at T0. Time-point percentages were plotted in Prism, and non-linear fit regression analysis was used to calculate statistics.

Time between metaphase and furrow ingression was taken on three points of imaging from a sample of 10 cells (chosen in different image quadrants) per video, with the number of frames counted between the stages (each representing 15 min). Time-0 was when the cells were in metaphase, as seen in brightfield by the chromosomal shadow along the metaphase plate, and counted until the first frame where a visible furrow was observed.

## Statistics

All statistical analyses were performed using GraphPad Prism Software (GraphPad). Student's *t* test and one-way ANOVA were used to determine significance unless otherwise noted. All error bars represent SEM unless otherwise noted. For all immunofluorescence experiments, images of a noted number of cells per biological replicate were taken for analysis. For quantitative immunofluorescence analysis of antibody signal, the same exposure was used for all images in that experiment and was quantified using FIJI. Data distribution was assumed to be normal, unless noted, where statistical tests were done to check distribution, in which case the median was used in place of the mean (Eddé et al, 1990; Glotzer, 2009; Guizetti et al, 2011; McNally et al, 2006; Vietri et al, 2015).

## Data availability

Source data for this study have been uploaded to BioStudies (S-BSST2212).

The source data of this paper are collected in the following database record: biostudies:S-SCDT-10_1038-S44319-026-00742-3.

## Peer review information

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

## Acknowledgements

This work was funded by the National Institutes of Health grant R01 GM143774 to RP, T32 GM136444-S3 to RS, and Bolie Scholar Award to RS. The authors declare no competing financial interests. We would like to acknowledge Dr. Carsten Janke for his and his lab at the Institute Curie's generous gift of many of the plasmids used in this study. We give additional acknowledgment and thanks to Dr. Kristen Verhey and her lab at the University of Michigan for letting us use their recombinant GT335 antibody. We also would like to acknowledge the Anschutz Microscopy core and the Barbra Davis

Center Sequencing core. Finally, we indebted to Dr. Jeffrey Moore (University of Colorado Anschutz Medical Campus) for all the expertise and advice during our work on microtubule dynamics and post-translational modifications.

## Author contributions

**Rachel Sachs**: Conceptualization; Resources; Data curation; Software; Formal analysis; Validation; Investigation; Visualization; Methodology; Writing—original draft; Writing—review and editing. **Yusuke Ogi**: Formal analysis; Validation. **Rytis Prekeris**: Conceptualization; Data curation; Formal analysis; Supervision; Funding acquisition; Investigation; Visualization; Methodology; Writing—original draft; Project administration; Writing—review and editing.

Source data underlying figure panels in this paper may have individual authorship assigned. Where available, figure panel/source data authorship is listed in the following database record: biostudies:S-SCDT-10_1038-S44319-026-00742-3.

## Disclosure and competing interests statement

The authors declare no competing interests.

