## [Peer Review File · EMBO Reports]

TPGS1 Regulates Central Spindle Microtubule Glutamylation and Remodeling During Telophase

Rachel Sachs, Yusuke Ogi, and Rytis Prekeris

Corresponding author(s): Rytis Prekeris (Rytis.Prekeris@ucdenver.edu)

Review Timeline:

Submission Date:	12th Jun 25
Editorial Decision:	7th Jul 25
Revision Received:	18th Sep 25
Editorial Decision:	24th Oct 25
Revision Received:	11th Nov 25
Accepted:	27th Nov 25

Editor: Deniz Senyilmaz Tiebe / Achim Breiling

Transaction Report:

Dear Rytis,

Thank you for submitting your manuscript to EMBO Reports, which was now seen by two referees, whose reports are copied below.

Referees express interest in the proposed role of tubulin polyglutamylated in regulation of telophase progression and abscission. However, they also raise some concerns that need to be addressed to consider publication here.

I find the reports informed and constructive, and believe that addressing the concerns raised will significantly strengthen the manuscript. As the reports are below, and I think all points need to be addressed, I will not detail them here. Please contact me if you have questions or comments regarding the revision for further discussion (also by video chat).

Should you be able to address all referee concerns, we would like to invite you to revise your manuscript with the understanding that the referee concerns (as in their reports) must be fully addressed and their suggestions taken on board. Please address all referee concerns in a complete point-by-point response. Acceptance of the manuscript will depend on a positive outcome of a second round of review. It is EMBO reports policy to allow a single round of major experimental revision only and acceptance or rejection of the manuscript will therefore depend on the completeness of your responses included in the next, final version of the manuscript.

We realize that it is difficult to revise to a specific deadline. In the interest of protecting the conceptual advance provided by the work, we recommend a revision within 3 months. Please discuss the revision progress ahead of this time with me if you require more time to complete the revisions, or if you have questions or comments regarding the revision (also by video chat).

1. A data availability section providing access to data deposited in public databases is missing (where applicable).
2. Your manuscript contains statistics and error bars based on $n=2$. Please use scatter plots in these cases.

You can submit the revision either as a Scientific Report or as a Research Article. For Scientific Reports, the revised manuscript can contain up to 5 main figures and 5 Expanded View figures, and it should not exceed 27000 characters. If the revision leads to a manuscript with more than 5 main figures it will be published as a Research Article. In this case the Results and Discussion section should be separate. If a Scientific Report is submitted, these sections have to be combined. This will help to shorten the manuscript text by eliminating some redundancy that is inevitable when discussing the same experiments twice. In either case, all materials and methods should be included in the main manuscript file.

4) a .docx formatted letter INCLUDING the reviewers' reports and your detailed point-by-point responses to their comments. As part of the EMBO publication's Transparent Editorial Process, EMBO reports publishes online a Review Process File (RPF) to accompany accepted manuscripts. This File will be published in conjunction with your paper and will include the referee reports,

your point-by-point response and all pertinent correspondence relating to the manuscript.

<https://www.embopress.org/page/journal/14693178/authorguide#transparentprocess>

5) a complete author checklist, which you can download from our author guidelines

<https://www.embopress.org/page/journal/14693178/authorguide>. Please insert information in the checklist that is also reflected in the manuscript. The completed author checklist will also be part of the RPF.

6) Please note that all corresponding authors are required to supply an ORCID ID for their name upon submission of a revised manuscript (<<https://orcid.org/>>). Please find instructions on how to link your ORCID ID to your account in our manuscript tracking system in our Author guidelines

<<https://www.embopress.org/page/journal/14693178/authorguide#authorshipguidelines>>

7) Before submitting your revision, primary datasets produced in this study need to be deposited in an appropriate public database (see <https://www.embopress.org/page/journal/14693178/authorguide#datadeposition>). Please remember to provide a reviewer password if the datasets are not yet public. The accession numbers and database should be listed in a formal "Data Availability" section placed after Materials & Method (see also

<https://www.embopress.org/page/journal/14693178/authorguide#datadeposition>). Please note that the Data Availability Section is restricted to new primary data that are part of this study. * Note - All links should resolve to a page where the data can be accessed. *

Additional information on source data and instruction on how to label the files are available:

<https://www.embopress.org/page/journal/14693178/authorguide#sourcedata>

9) Our journal encourages inclusion of *data citations in the reference list* to directly cite datasets that were re-used and obtained from public databases. Data citations in the article text are distinct from normal bibliographical citations and should directly link to the database records from which the data can be accessed. In the main text, data citations are formatted as follows: "Data ref: Smith et al, 2001" or "Data ref: NCBI Sequence Read Archive PRJNA342805, 2017". In the Reference list, data citations must be labeled with "[DATASET]". A data reference must provide the database name, accession number/identifiers and a resolvable link to the landing page from which the data can be accessed at the end of the reference. Further instructions are available at <http://www.embopress.org/page/journal/14693178/authorguide#referencesformat>

10) Regarding data quantification (see Figure Legends:

<https://www.embopress.org/page/journal/14693178/authorguide#figureformat>)

12) Please also note our reference format:

13) All Materials and Methods need to be described in the main text using our 'Structured Methods' format, which is required for all research articles. According to this format, the Methods section includes a Reagents and Tools Table (listing key reagents, experimental models, software and relevant equipment and including their sources and relevant identifiers) followed by a Methods and Protocols section describing the methods using a step-by-step protocol format. The aim is to facilitate adoption of the methodologies across labs. More information on how to adhere to this format as well as a downloadable template (.docx) for the Reagents and Tools Table can be found in our author guidelines:

I look forward to seeing a revised version of your manuscript when it is ready. Please let me know if you have questions or comments regarding the revision.

Kind regards,

Deniz

Deniz Senyilmaz Tiebe, PhD
Senior Scientific Editor
EMBO Reports

Referee #1:

The manuscript by Sachs and colleagues investigates the role of tubulin glutamylation during the final phases of cell division. Glutamylation is a posttranslational modification that occurs on both a-tubulin and b-tubulin subunits and is thought to impact the binding of specific MAPs and/or motors. The authors use immunofluorescence to demonstrate changes in glutamylation over the stages of mitosis. They determine that the glutamylation enzymes TLL1 and TLL11 localize to the midbody. And they show that loss of TLL1 via knock out of its localization partner TPGS1 results in a delay in abscission. The data are well-presented and the quantification is rigorous. The schematics are well done and helpful, especially for understanding the ICB width and aspect ratio calculations. I had some trouble following the story so the writing could use some streamlining and editing for clarity. And there are some major concerns that need to be addressed as well as some minor presentation issues.

Major comments:

1. The paper is lacking information on all antibodies used including catalog numbers and concentrations.
2. Please do not use red/green color scheme for the images, superplots, or schematics.
3. Figures 2 and 3 aim to demonstrate changes in glutamylation over the stages of mitosis. But Figure 3 appears to be a repeat of Figure 2 with additional stages yet fewer antibodies. Can these two figures be merged and/or simplified?
 - a. Assuming the a-tubulin antibody is DM1a, have the authors done single antibody immunofluorescence experiments to exclude the possibility that there are negative (steric and/or competitive) interactions with these antibodies in the case that they recognize neighboring epitopes?
 - b. Assuming the PolyE antibody is IN105, can the authors comment on why their staining looks so different from HeLa cell staining in the literature (Zadra 2022)? Is this PolyE antibody truly working?
 - c. How does the polyE change in anaphase and furrow ingression?
 - d. How was the quantification done in Figure 2? No a-tubulin staining is shown yet the quantification appears to be relative to a-tubulin. Is the quantification for rGT335 relative to acetylated a-tubulin?
 - e. Can the authors comment on why the rGT335 staining clearly decorates the metaphase spindle in Figure 2 but not in Figure 3?
 - f. In Figure 3 panel B, the quantification shows and the authors state that there is low colocalization between rGT335 and a-tubulin staining in anaphase. But based on the images in panel A, it looks like the highest rGT335 signal and the highest

colocalization of rGT335 and α -tubulin is during anaphase.

4. Figure 5 introduces the TPGS1 KO cell line with respect to TLL1 localization. Does expression of TPGS1 rescue TLL1 localization in the TPGS1 KO? This line is later characterized in Figure 7. Perhaps the authors could consider reorganizing Figures 5-7 to make the story flow better.

5. For characterization of the TPGS1 KO cell line, the y-axis label "Percent of cells post mitosis" in Figure 7A is confusing. Is this the time to complete mitosis? Does this exclude cells that never completed mitosis? From the methods description it sounds more like this is the number of cells that complete cytokinesis during the observed period. Are other aspects of mitosis changed, for example the time from NEB to metaphase or the duration of mitosis? Is there an increase in binucleate cells in the KO cells? The stated delay in abscission needs to be quantified.

6. Figure 6 is missing important controls. First, they need to show that the TPGS1 3'UTR drives mRNA localization to the midbody. Second, they need to show that a different (control) 3'UTR does not cause TPGS1-EYFP protein to localize to the midbody.

7. The authors consider three ways that TPGS1/TLL1 KO could be affecting abscission. To examine whether there is an effect on crosslinking of ICB microtubules, they stain for PRC1. The authors state that "PRC1 intensity within the ICB remained unchanged." However, the data in Supp Figure 9 suggest that PRC1 localization is changed and this needs to be quantified.

8. The authors then consider whether TPGS1/TLL1 KO could be affecting microtubule severing. Figure 8G shows that severing is decreased in TPGS1 KO cells. Does expression of TPGS1 in KO cells rescue the percentage of late telophase cells with severed microtubules? Does overexpression of TPGS1 in control cells drive increased levels of severing?

9. In Figure 9 and the discussion, the authors discuss a model that pulls together their data and the literature. However, there are several discrepancies with the literature. Contrary to what is stated by the authors (p.3), spastin is regulated by longer chain glutamylation (La Croix 2010, Valenstein and Roll-Mecak 2016) with peak function at 5-8E. Furthermore, TLL1 preferentially modifies α -tubulin (Janke 2005, Bodakuntla 2021), but spastin only requires the β -tubulin tail (Valenstein and Roll-Mecak 2016). Finally, spastin localization is unchanged in TPGS1-KO cells (Figure 7) and does not seem to phenocopy spastin KO HeLa cells as previously reported by the Prekeris lab (Schiel 2011). Please address these issues.

10. In the discussion, the authors propose a model in which spindle glutamylation is removed during anaphase and then added back in telophase. As noted in the discussion, CCP1 and 5 preferentially act on free tubulin subunits, but MT stabilization and compaction of the central spindle begins during anaphase (do Rosario 2023, Asthana 2021), which would be at odds with this model. Is there necessarily enrichment of GT335 on the central spindle that needs to be removed? The metaphase image in Figure 3 does not appear to have much GT335 signal beyond the spindle poles, although the image in Figure 2 might have glutamylated bridging fibers

Minor comments:

11. What specific HeLa cells are being used?

12. The overall timing of the localization of glutamylation and the TLL enzymes is confusing. For example, in figure 3, the rGT335 signal appears to be already enriched by early telophase but TLL1 does not appear to be enriched until late telophase and TLL11 localizes to the midbody (Figure 4)? Can the authors explain these discrepancies? It would be helpful for the reader unfamiliar with the details for the authors to explain the enzymatic activities of these enzymes.

13. It would also be helpful for the authors to walk the reader through what the antibodies are detecting. For example, Figure 8 indicates that rGT335 staining is lost at the midbody in the TPGS1 KO cells whereas Supp Figure 7 shows no change in polyE. How can this be if rGT335 is marking the first (branching) glutamate and polyE is marking the glutamates extended off of the branching glutamate?

14. Figure 6 would benefit from quantification of TPGS1 localization. For example, it is very difficult to see the localization to the MB in Figure 6C.

15. Figure 7 shows that the minus ends of the ICB microtubules have a "fan out" phenotype in the TPGS1 KO cells. TPGS1 seems to localize to the minus ends of these MTs (Figure 6). Can the localization of TPGS1 be quantified? How is the ICB splitting described in Fig 7G,H different from the fan-out of the ICB?

16. Since spastin localization is unchanged despite the decrease in severing, can the authors examine katanin localization?

17. In Figure 8D,E both visually and by quantification it looks like there is a decrease in GT335 signal in the TPGS1 KO cells, although in the example image there also appears to be less α tubulin staining in the TPGS1 KO image. Can the authors comment on this or break out the graph into rGT335 and α -tubulin intensity?

18. The work on mRNA localization at midbody should mention Park 2023 and discuss whether the two studies agree on TPGS1 localization.

19. The western blot in Figure 5 is very large.

20. Typos/grammatical errors:

--p.4 "the nucleus is reniform initially"

--p.5 "Consistent with previous studies" requires references

--p.5 "progression from metaphase to telophase involves increase in short chain glutamylation." This is missing an "an" between involves and increase.

--p.5 "anti-rGT335 antibody". Anti-rGT335 indicates that rGT335 is the epitope.

--p.8 ""cells to undergo abscission cells"

Referee #2:

In this manuscript the authors examine microtubule polyglutamylation, a tubulin post-translational modification, during the final stages of mitosis and abscission. The main findings are that midzone/intracellular bridge microtubules acquire short-chain polyglutamylation during telophase. Cells lacking TPGS1, which is part of the complex involved in targeting the polyglutamylase to microtubules, show defects in abscission and organization of ICB microtubules. Overall, the question(s) are of interest, but issues with various aspects of experimental design/data analysis and resulting conclusions need to be addressed prior to publication.

1. Short chain vs long chain polyglutamylation: antibody staining was used to document the existence of these post-translational modifications on microtubules in late mitosis and abscission. The polyE antibody staining appears to be non-specific as it appears as 'dots' in the cytoplasm (Fig2; see also comment below regarding supplemental data). Perhaps a positive control showing on cilia or flagella might be shown to document what actual microtubule staining looks like with this antibody; in any case, the data shown is not convincing. I suggest this be eliminated and the text reflect the fact that no specific staining was observed.

Regarding data showing staining with the rGT335 antibody- please include a panel with a stage between Metaphase and Telophase in Figure 2 so that the presumed lack of staining prior to telophase can be seen. For figure 1, showing the intracellular bridge microtubules in early and late telophase, use the same magnification of the boxed regions so a more direct comparison of the morphology of these microtubules can be appreciated (ICB microtubules are getting shorter?). The co-localization data seems like it could be omitted (polyE staining) or moved to supplemental - why quantify the non-specific dots in the anaphase and ingressing cells? Figure 3 - Figures were not numbered in the pdf, and that would have made reading much easier.

2. Figure 4 shows the localization of overexpressed TTLL1 at the very center of the ICB microtubules, presumably the region of microtubule overlap. The signal from rGT335, showing where microtubules are post-translationally modified, is all along the ICB microtubules. Why is the modifying enzyme restricted in localization (central overlap) whereas the modification is all along the ICB microtubules? (the expressed protein is observed at overlap whereas antibody signal is not - presumably due to antibody exclusion by the highly proteinaceous and compacted mid-body - this should be mentioned as well).

3. To explore the possibility that the post-translational modification may impact the process of cell division and/or abscission the authors generate a KO cell line lacking the targeting factor for TTLL1; control cells localize TTLL1 whereas the KO cells do not indicating that the TPGS is needed to localize TTLL1. The signal of TPGS in live cells is extremely dim - comment? And most of the expressed protein is present as bright puncta near the centrosome - comment? Does the localization of punctae depend on microtubules? actin? The authors show (Fig 6 D) cells expressing both the TTLL1 and TPGS - all that I can see is blobs of various sizes and no obvious structure or co-localization. This should be omitted, repeated or explained. It is possible that these are low abundance proteins, and the blobs are not specific.

4. In Figure 7,8 cells KO of TPGS are examined. The data suggest that there are changes or defects in late mitosis and abscission, but the experiments need to be performed so that the timing of the cells is the same. For example, the authors use live cell imaging to call attention to changes in midzone-ICB microtubules. However, the timing needs to be more carefully controlled. In Figure 7 B the control cell movie begins ("0" min) when the icb is compact and short; the TPGS KO cell movie begins ("0" min) at an (apparently) earlier stage when the midzone has not shortened and is still a "bow tie". Why not use AO as time zero? Again in Fig 7 C, the top row (control) begins with a very short midzone/ICB. The KO cell starts with a longer midzone/ICB and the nuclei (judging from position of centrosome microtubules) has not yet flattened (the criteria for staging cells mentioned by the authors). To be clear, the data suggest that the cells are, in fact, defective as evidenced by the bent microtubules in ICB of KO cells and the failure to shorten the microtubules over 90 min (panel B). The data will be much more convincing if the movies all began at AO. In panel D of this figure the cells are fixed, so getting comparable stages is more difficult, but the control cell looks like a later stage than the KO. The splitting of the microtubules at the ends of the ICB is marked in the acetylated tub image, not the alpha tubulin; should not the microtubules in both show the splitting? This seems like it needs to be addressed. Could the apparent splitting be interphase microtubules in the daughter cells

that overlap with the minus ends of ICB microtubules? Most interestingly, in the KO cell in G, lower, there appear to be 2 cut sites, one on either side of the asterisk. Is this observed in the KO cells? Please comment.

5. in figure 8 data on abscission is shown with images for cut and not-cut control cells. Where are the images of the KO cells? Additionally the control cells show very nice splitting of the microtubules f the ICB (example 2)! How frequent is this organization observed?

Other

Page 5 top. Add a reference for no stability of ICB microtubules.

For mean intensities where was the background measured, esp in cases where non specific dots were observed.

Supplemental figure 3. Why is there no alpha tubulin staining in the midzone (second image)? Why is the mask for the fourth image only a small circle in center - why not use all the midbody microtubule area?

Supplemental figure 4. There is little or no midbody/ICB signal with the PolyE antibody. To examine role of CCP1 and 5, why not use rGT335 staining? Why would CCP 1 or 5 be expected to label nuclei? Presumably the YFP is small enough to enter nucleus non specifically.

Figure 2 B shows punctate staining with PolyE that looks non specific; Figure Suppl 7 shows some (dim or blurry) staining of ICB with Poly E. What is going on with the data - does this antibody reliably identify Poly E (glutamylation) on the MB or ICB microtubules? In Suppl Fig 7 why the bright intensity near nucleus (top row control cell) and the bright blob in KO cell second row?

Supplemental figure 8, panel A, KO cell shows signal at MB; control does not. Please comment.

Supplemental Figure 9 shows PRC staining (panel A) which is expected to localize at regions of antiparallel microtubule overlap. Surprisingly in the KO cells PRC is located distal to the zone of overlap in the region of the 'splay-out'. Please comment. Also for the spastin, images are shown for control cells; please include KO cells. For the control cell bottom row multiple dots of spastin are observed; are there two cut sites? Were both dots of spastin quantified? Are multiple dots of spastin staining frequently observed?

We would like to thank the Reviewers for very constructive comments and suggestions. In this revised manuscript we incorporated most of them and we believe that this enhanced the manuscript. The point-by-point rebuttal is listed below. All text changes in manuscript are marked in yellow.

Referee #1:

The manuscript by Sachs and colleagues investigates the role of tubulin glutamylation during the final phases of cell division. Glutamylation is a posttranslational modification that occurs on both α -tubulin and β -tubulin subunits and is thought to impact the binding of specific MAPs and/or motors. The authors use immunofluorescence to demonstrate changes in glutamylation over the stages of mitosis. They determine that the glutamylation enzymes TTLL1 and TTLL11 localize to the midbody. And they show that loss of TTLL1 via knock out of its localization partner TPGS1 results in a delay in abscission. The data are well-presented and the quantification is rigorous. The schematics are well done and helpful, especially for understanding the ICB width and aspect ratio calculations. I had some trouble following the story so the writing could use some streamlining and editing for clarity. And there are some major concerns that need to be addressed as well as some minor presentation issues.

Major comments:

1. The paper is lacking information on all antibodies used including catalog numbers and concentrations.

We have added all antibodies to the method section.

2. Please do not use red/green color scheme for the images, superplots, or schematics.

As suggested, we have changed color scheme.

3. Figures 2 and 3 aim to demonstrate changes in glutamylation over the stages of mitosis. But Figure 3 appears to be a repeat of Figure 2 with additional stages yet fewer antibodies. Can these two figures be merged and/or simplified?

As suggested by reviewers we have moved PolyE data to a supplemental figure, which allowed us to combine Figure 2 and Figure 3. Former Figure 3 (now Figure 2D-E) aims to look specifically at the glutamylation of central spindle microtubules that become the ICB in telophase.

In previous Figure 3 (now Figure 2D-E) we only stained with rGT335 antibodies since in Figure 2 we demonstrate that long-chain polyglutamylation, that is detected with PolyE antibody, is not present in ICB of telophase cells. As suggested, we have now added anaphase cell images stained for PolyE antibody which are now shown in Supplemental Figure 2.

a. Assuming the α -tubulin antibody is DM1a, have the authors done single antibody immunofluorescence experiments to exclude the possibility that there are negative (steric and/or

competitive) interactions with these antibodies in the case that they recognize neighboring epitopes?

Yes, we used the DM1a antibody, thus, we fully agree with the reviewer that there is a possibility for steric and competitive issues. To control for that possibility in this revised manuscript we co-stained rGT335 and PolyE with anti-acetylated α -tubulin antibody that should not cause any steric or competitive issues. The data is shown in Supplemental Figure 2A (for rGT335) and Supplemental Figure 3B (for PolyE). Importantly, the data from new staining anti-acetylated α -tubulin antibody is fully consistent with data obtained co-staining with DM1a.

b. Assuming the PolyE antibody is IN105, can the authors comment on why their staining looks so different from HeLa cell staining in the literature (Zadra 2022)? Is this PolyE antibody truly working?

Yes, PolyE is IN105. We also agree with the reviewer that it is not the best antibody since it gives quite a bit of background staining. However, the antibody does work since in metaphase we see very clear (and above the background) signal that colocalizes with mitotic spindle. This specific signal disappears in telophase cells, consistent with our model that high chain polyglutamylation is absent when the cell progresses from metaphase to telophase. We replaced the PolyE images in previous Figure 2 (now Supplemental Figure 3) with the ones that are better representation of what we see. We also tested PolyE antibody in ciliated cells to further confirm its validity. PolyE clearly stains cilia (see image below) further confirming that antibody does work for immunofluorescence.

c. How does the polyE change in anaphase and furrow ingression?

As suggested, we added polyE images of anaphase and ingressing cells to Supplemental Figure 3E. Consistent with our model that long-chain glutamylation is removed after metaphase, we do not see any specific polyE signal in anaphase and ingressing cells when looking at central spindle microtubules.

d. How was the quantification done in Figure 2? No a-tubulin staining is shown yet the

quantification appears to be relative to α -tubulin. Is the quantification for rGT335 relative to acetylated α -tubulin?

We have added clarification to the methods section. Images in figure 2 were taken on confocal co-stained with acetylated α -tubulin to get the best possible initial images. As the reviewer noted earlier, we shared some concerns about α -tubulin interference. Additionally, our acetylated tubulin antibody is much stronger and provides a more detailed visual of the ICB microtubules, making it better for visuals. However, we wanted to assess modification levels relative to tubulin amount to attempt to account for differences in tubulin levels that might otherwise skew the data. All analysis reflected in the graphs was done on a separate set of non-confocal images where antibodies were co-stained with α -tubulin.

e. Can the authors comment on why the rGT335 staining clearly decorates the metaphase spindle in Figure 2 but not in Figure 3?

We replaced the image that is better representative of rGT335 staining in metaphase.

f. In Figure 3 panel B, the quantification shows and the authors state that there is low colocalization between rGT335 and α -tubulin staining in anaphase. But based on the images in panel A, it looks like the highest rGT335 signal and the highest colocalization of rGT335 and α -tubulin is during anaphase.

rGT335 does colocalize quite strongly with kinetochore microtubules in both metaphase and anaphase. In new Figure 2 (used to be Figure 3), our goal was to attempt to measure the rGT335 signal on central spindle microtubules (localized in center of the cell). To minimize the confusion, we edited manuscript to make that more clear. We also re-did imaging and quantification (see Figure 2) of rGT335 signal on central spindle microtubules.

4. Figure 5 introduces the TPGS1 KO cell line with respect to TTLL1 localization. Does expression of TPGS1 rescue TTLL1 localization in the TPGS1 KO? This line is later characterized in Figure 7. Perhaps the authors could consider reorganizing Figures 5-7 to make the story flow better.

Based on reviewer comments and new data from revisions we have rearranged these figures. Unfortunately, cells do not tolerate TPGS1 or TTLL1 co-overexpression. However, as suggested, we did do a rescue experiment with adding back TPGS1 in TPGS1-KO cells to see whether it rescues microtubule phenotype. The data is now shown in Figure 6F-G.

5. For characterization of the TPGS1 KO cell line, the y-axis label "Percent of cells post mitosis" in Figure 7A is confusing. Is this the time to complete mitosis? Does this exclude cells that never completed mitosis? From the methods description it sounds more like this is the number of cells that complete cytokinesis during the observed period. Are other aspects of mitosis changed, for example the time from NEB to metaphase or the duration of mitosis? Is there an increase in binucleate cells in the KO cells? The stated delay in abscission needs to be quantified.

Yes, we measured the time to complete mitosis, thus, the measurement excludes the cells that did not complete mitosis. We have edited the methods section to make the explanation of the analysis better. Unfortunately, we cannot measure the timing from NEB to mitosis since these cells were imaged using bright-field microscopy and we do not have a good way of determining NEB timing. However, as suggested we have investigated time needed for cells to transition between metaphase through anaphase to furrow ingression. As shown in the Figure 5D, there was no difference in metaphase-anaphase transition time between control and TPGS1-KO cells, suggesting that delay in cell division likely comes from delay in telophase-abscission.

6. Figure 6 is missing important controls. First, they need to show that the TPGS1 3'UTR drives mRNA localization to the midbody. Second, they need to show that a different (control) 3'UTR does not cause TPGS1-EYFP protein to localize to the midbody.

We agree with the reviewer that it would be interesting to test whether TPGS1 3'UTR is necessary and sufficient to drive mRNA localization to the midbody. However, mRNA targeting is not a focus of this manuscript (although we are currently also working on it for different publication). It would also require quite a bit of additional experiments to complete this type of study. We included the 3'UTR out of caution based on some similarities to other mRNA reported in our previous publication (Farmer and Vaeth et al. 2023). Consequently, we believe that these types of studies are outside the scope of this manuscript.

7. The authors consider three ways that TPGS1/TLL1 KO could be affecting abscission. To examine whether there is an effect on crosslinking of ICB microtubules, they stain for PRC1. The authors state that "PRC1 intensity within the ICB remained unchanged." However, the data in Supp Figure 9 suggest that PRC1 localization is changed and this needs to be quantified.

We have quantified PRC1 signal at the centrals spindle in wild-type and KO cells and observed no difference (see Supplemental Figure 8B). PRC1 staining may look a bit different since KO cells have longer and splayed out microtubules. However, quantification of PRC1 signal/per area shows that PRC1 signal on microtubules is not different between wild-type and KO cells.

8. The authors then consider whether TPGS1/TLL1 KO could be affecting microtubules. TPGS1 in KO cells rescue the percentage of late telophase cells with severed microtubules? Does overexpression of TPGS1 in control cells drive increased levels of severing?

We do agree that this would be an interesting experiment to do. However, cells really do not like over-expressing TPGS1 or TLL1, presumably because it affects microtubule function. That has been reported for other TLL glutamylases as well. This is one of the reasons why we had to transfect cells with TLL enzymatic mutant to analyze its localization. Consequently, due to technical issues we cannot perform the proposed experiment. We did, however, do a TPGS1 rescue experiment and show that over-expression of TPGS1 partially rescues the KO phenotype. (Figure 6F-G).

9. In Figure 9 and the discussion, the authors discuss a model that pulls together their data and the literature. However, there are several discrepancies with the literature. Contrary to what is stated by the authors (p.3), spastin is regulated by longer chain glutamylation (La Croix 2010,

Valenstein and Roll-Mecak 2016) with peak function at 5-8E. Furthermore, TTLL1 preferentially modifies a-tubulin (Janke 2005, Bodakuntla 2021), but spastin only requires the b-tubulin tail (Valenstein and Roll-Mecak 2016). Finally, spastin localization is unchanged in TPGS1-KO cells (Figure 7) and does not seem to phenocopy spastin KO HeLa cells as previously reported by the Prekeris lab (Schiel 2011). Please address these issues.

We fully agree with the reviewer that there is still much to be learned about how TPGS1 functions to mediate central spindle microtubule re-modeling during telophase. Spastin appears to be one of the candidates, but we certainly cannot rule out that there are other microtubule regulators that are affected by lack of TPGS1-induced tubulin glutamylation. It is also important to note that while Valenstein and Roll-Mecak did show that 5-8E is optimal to regulate Spastin enzymatic function, all these experiments were done in vitro. The same applies for apparent requirement of only b-tails for Spastin to functions. It will be important in the future to demonstrate that the same rules also apply for Spastin in much more complex in cello (or in vivo) context.

Our finding that TPGS1 KO appears to affect Spastin cutting but not recruitment to the central spindle microtubules is actually not that surprising since it was demonstrated by few labs that ESCRT-III complex, rather than microtubules, facilitates Spastin localization to the abscission site.

We feel that TPGS1 KO does phenocopy quite well the Spastin KD that we published several years ago (Schiel 2011). In that study we have shown that Spastin depletion leads to increased disorganization (and decreased anti-parallel bundling) and length of central spindle microtubules, the phenotype we also observe in TPGS1 KO (see Figure 5-6). That is another line of evidence implicating that Spastin contributes to TPGS1-KO induced phenotypes during telophase.

As suggested, we have expanded our Discussion section to address/explore all these issues.

10. In the discussion, the authors propose a model in which spindle glutamylation is removed during anaphase and then added back in telophase. As noted in the discussion, CCP1 and 5 preferentially act on free tubulin subunits, but MT stabilization and compaction of the central spindle begins during anaphase (do Rosario 2023, Asthana 2021), which would be at odds with this model. Is there necessarily enrichment of GT335 on the central spindle that needs to be removed? The metaphase image in Figure 3 does not appear to have much GT335 signal beyond the spindle poles, although the image in Figure 2 might have glutamylated bridging fibers

We fully agree with the reviewer and also think it's unlikely that glutamylation is removed during anaphase (although we cannot fully discount this possibility). Our new data appears to show that central spindle microtubules are likely not glutamylated during metaphase and anaphase, unlike kinetochore microtubules. Consequently, we hypothesize that glutamylation is added to central spindle microtubules during telophase in TPGS1-dependent fashion.

We edited the Discussion section to make that more clear.

Minor comments:

11. What specific HeLa cells are being used?

All HeLa cells used in these experiments were purchased from ATCC. We added that information to the Method section.

12. The overall timing of the localization of glutamylation and the TLL enzymes is confusing. For example, in figure 3, the rGT335 signal appears to be already enriched by early telophase but TLL1 does not appear to be enriched until late telophase and TLL1 localizes to the midbody (Figure 4)? Can the authors explain these discrepancies? It would be helpful for the reader unfamiliar with the details for the authors to explain the enzymatic activities of these enzymes.

This is one of more interesting (and confusing) findings in our studies. While TPGS1 and TLL1 are both localized at the midbody, there are also clear differences in their localization. Specifically, TPGS1 can clearly be observed at the minus ends of central spindle microtubules even without presence of TLL1. We are not fully sure what that means. It could be that TPGS1 may also form complex with other TLLs. Alternatively, TPGS1-based targeting complex may form first, and then recruit TLL1 later. We are definitely interested in dissecting these possibilities further, but that is outside the scope of this study. We added discussion about these possibilities in Discussion section.

13. It would also be helpful for the authors to walk the reader through what the antibodies are detecting. For example, Figure 8 indicates that rGT335 staining is lost at the midbody in the TPGS1 KO cells whereas Supp Figure 7 shows no change in polyE. How can this be if rGT335 is marking the first (branching) glutamate and polyE is marking the glutamates extended off of the branching glutamate?

Very valid observation by reviewer. Since, upon further investigation in quality of polyE (see rebuttal to comment 3b), it is clear that polyE signal during telophase is a background staining, we have removed original Supplemental Figure 7 from the manuscript.

14. Figure 6 would benefit from quantification of TPGS1 localization. For example, it is very difficult to see the localization to the MB in Figure 6C.

As suggested, we added quantification of TPGS1 localization during telophase (see Figure 4E).

15. Figure 7 shows that the minus ends of the ICB microtubules have a "fan out" phenotype in the TPGS1 KO cells. TPGS1 seems to localize to the minus ends of these MTs (Figure 6). Can the localization of TPGS1 be quantified? How is the ICB splitting described in Fig 7G,H different from the fan-out of the ICB?

We have added detailed quantification of TPGS1 localization (Figure 4E). As for "splitting", it is not that different for "fan out". We simply use splitting as another way of measuring defects in central spindle microtubule organization. Splitting represents a type of fan-out where, as the nucleus reforms, the microtubules have to curve around it if it is in conflict with the

microtubules. In control cells, the minus ends have been reorganized and bundled in such away where this conflict is less likely to occur.

16. Since spastin localization is unchanged despite the decrease in severing, can the authors examine katanin localization?

We have examined katanin localization in mitotic cells and added that data to Supplemental Figure 10. Katanin is mostly present at the centrosomes (previously shown) with low levels of katanin also present ICB minus ends in both early and late telophase, and that remains the case in the TPGS1-KO cells.

17. In Figure 8D, E both visually and by quantification it looks like there is a decrease in GT335 signal in the TPGS1 KO cells, although in the example image there also appears to be less alpha tubulin staining in the TPGS1 KO image. Can the authors comment on this or break out the graph into rGT335 and a-tubulin intensity?

In all quantifications (see Figure 7C-E) we normalize glutamylation signal to tubulin signal to compensate for variability in tubulin levels. As suggested, we also included separated graphs of rGT335 and a-tubulin intensities (see Supplemental Figure 6). Somewhat surprisingly despite the appearance by eye, a-tubulin signal is higher in TPGS1-KO cells than in controls. We suspect this is in line with decreases in remodeling of microtubules. As the rGT335 antibody is weaker the quantification shows a more subtle decrease in intensity in rGT335 before a-tubulin is taken into account. Meanwhile the a-tubulin antibody is much stronger and so despite a greater change it is less visible by eye.

18. The work on mRNA localization at midbody should mention Park 2023 and discuss whether the two studies agree on TPGS1 localization.

We looked at RNAseq data shown in Park 2023. The data does not list TPGS1 mRNA as enriched at the midbody. Please note, however, that Park 2023 only has one replicate of RNAseq analysis. Additionally, they used very different method of purifying MBs. That makes difficult to compare our data sets. As suggested, we added mention of Park 2023 study to our Discussion section.

19. The western blot in Figure 5 is very large.

We have decreased the size of the blot to fit better into the figure.

20. Typos/grammatical errors:

--p.4 "the nucleus is reniform initially"

--p.5 "Consistent with previous studies" requires references

--p.5 "progression from metaphase to telophase involves increase in short chain glutamylation."

This is missing an "an" between involves and increase.

--p.5 "anti-rGT335 antibody". Anti-rGT335 indicates that rGT335 is the epitope.

--p.8 ""cells to undergo abscission cells"

We have revised the noted errors.

Referee #2:

In this manuscript the authors examine microtubule polyglutamylation, a tubulin post-translational modification, during the final stages of mitosis and abscission. The main findings are that midzone/intracellular bridge microtubules acquire short-chain polyglutamylation during telophase. Cells lacking TPGS1, which is part of the complex involved in targeting the polyglutamylase to microtubules, show defects in abscission and organization of ICB microtubules. Overall, the question(s) are of interest, but issues with various aspects of experimental design/data analysis and resulting conclusions need to be addressed prior to publication.

1. Short chain vs long chain polyglutamylation: antibody staining was used to document the existence of these post-translational modifications on microtubules in late mitosis and abscission. The polyE antibody staining appears to be non-specific as it appears as 'dots' in the cytoplasm (Fig2; see also comment below regarding supplemental data). Perhaps a positive control showing on cilia or flagella might be shown to document what actual microtubule staining looks like with this antibody; in any case, the data shown is not convincing. I suggest this be eliminated and the text reflect the fact that no specific staining was observed.

In response to reviewer suggestions, we have further tested the polyE antibody. It clearly stains cilia (as reported by other labs) as shown in the image below. We also re-imaged polyE staining in dividing cells (and replaced images in Figure 2/now supplement 3). It is very clear that polyE stains the mitotic spindle in metaphase (see new image in Figure 3) but that staining goes away in telophase. Thus, reviewer is correct, any signal seen in telophase is likely a background staining. We have edited manuscript text to make that clear.

Regarding data showing staining with the rGT335 antibody- please include a panel with a stage between Metaphase and Telophase in Figure 2 so that the presumed lack of staining prior to telophase can be seen. For figure 1, showing the intracellular bridge microtubules in early and late telophase, use the same magnification of the boxed regions so a more direct comparison of the morphology of these microtubules can be appreciated (ICB microtubules are getting

shorter?).

The co-localization data seems like it could be omitted (polyE staining) or moved to supplemental - why quantify the non-specific dots in the anaphase and ingressing cells? Figure 3 - Figures were not numbered in the pdf, and that would have made reading much easier.

As suggested, we made several changes to figures 1-3. Specifically: 1) We added boxed regions in figure 1 of the same magnification; 2) Numbered figures; 3) Moved polyE quantification to supplemental figure 3; 4) Added anaphase and ingressing cell images stained with rGT335 to Figure 2; 5) Revised quantification in figure 3 (now figure 2) to measure furrow ingression and early telophase only as they are the stages we can ascertain the central spindle microtubule population clearly.

2. Figure 4 shows the localization of overexpressed TTLL1 at the very center of the ICB microtubules, presumably the region of microtubule overlap. The signal from rGT335, showing where microtubules are post-translationally modified, is all along the ICB microtubules. Why is the modifying enzyme is restricted in localization (central overlap) whereas the modification is all along the ICB microtubules? (the expressed protein is observed at overlap whereas antibody signal is not - presumably due to antibody exclusion by the highly proteinaceous and compacted mid-body - this should be mentioned as well).

We agree that the localization of TTLL1 and glutamylation in central spindle is interesting and still somewhat puzzling. Unfortunately, we do not have a clear explanation about it. There could be many reasons for that. One, TTLL1 signal may just be too low to be observed on central spindle microtubules, while it can be observed in the midbody where there are more microtubules. TTLL1 is typically expressed at much lower levels compared to TPGS1. Two, TTLL1 may actually function early in telophase while central spindle microtubules are being compacted and no clear midbody has formed yet. Thus, the appearance of TTLL1 in the midbody during late telophase may be a result of central spindle compacting and shortening (thus concentrating TTLL1 in the midbody). We are interested in investigating these possibilities further, but that would be outside the scope of this manuscript. You are correct about antibody exclusion in the midbody, we added mention of this in the methods section.

3. To explore the possibility that the post-translational modification may impact the process of cell division and/or abscission the authors generate a KO cell line lacking the targeting factor for TTLL1; control cells localize TTLL1 whereas the KO cells do not indicating that the TPGS is needed to localize TTLL1. The signal of TPGS in live cells is extremely dim - comment? And most of the expressed protein is present as bright puncta near the centrosome - comment? Does the localization of punctae depend on microtubules? actin? The authors show (Fig 6 D) cells expressing both the TTLL1 and TPGS - all that I can see is blobs of various sizes and no obvious structure or co-localization. This should be omitted, repeated or explained. It is possible that these are low abundance proteins, and the blobs are not specific.

Generally, cells really do not like to overexpress either TTLL1 or TPGS1 (and especially both). Presumably that is because it affects microtubule function. We have tried to generate stable cell lines over-expressing TTLL1 or TPGS1 but found that cells rapidly down-regulate GFP-TTLL1

or GFP-TPGS1 expression. That has been observed with other TLL glutamylases as well, thus, that is always a big issue in the field. Because of that, for imaging we always picked transiently transfected low-expressing cells to minimize the effect of overexpression on mitosis. Consequently, TPGS1 and TLL1 images are always dim.

We agree with the reviewer that images of TPGS1/TLL1 co-expressing cells are hard to interpret and we certainly cannot rule out the possibility that these are non-specific aggregations. As the result, we removed those images from the manuscript.

4. In Figure 7,8 cells KO of TPGS are examined. The data suggest that there are changes or defects in late mitosis and abscission, but the experiments need to be performed so that the timing of the cells is the same. For example, the authors use live cell imaging to call attention to changes in midzone-ICB microtubules. However, the timing needs to be more carefully controlled. In Figure 7 B the control cell movie begins ("0" min) when the icb is compact and short; the TPGS KO cell movie begins ("0" min) at an (apparently) earlier stage when the midzone has not shortened and is still a "bow tie". Again in Fig 7 C, the top row (control) begins with a very short midzone/ICB. The KO cell starts with a longer midzone/ICB and the nuclei (judging from position of centrosome microtubules) has not yet flattened (the criteria for staging cells mentioned by the authors). To be clear, the data suggest that the cells are, in fact, defective as evidenced by the bent microtubules in ICB of KO cells and the failure to shorten the microtubules over 90 min (panel B).

We have re-done time-lapse imaging experiments (see Figure 5E). In all cases we start imaging cells at metaphase and continued imaging cells (with 10 min time-lapse) until late telophase. Since cells can stay in metaphase for varying amounts of time, we defined as time-0 when cells started ingressing. This ensures that cells are all at the same stage. Importantly, data now clearly support our hypothesis that TPGS1 is needed for re-organization and compaction of central spindle microtubules.

In panel D of this figure the cells are fixed, so getting comparable stages is more difficult, but the control cell looks like a later stage than the KO.

As reviewer points out, correctly staging cells is a key for this type of analysis. We cannot really stage them based on length of the ICB since it varies quite a bit between cells. Additionally, we are knocking out microtubule modifying enzyme, thus, the shape and length of the ICB is itself effected by KO. Consequently, for this analysis we stage telophase cells using two major criteria: 1) extent of cell flattening (the flatter the cell the later in telophase cell is), and 2) roundness and size of the nucleus (the rounder the nucleus the later in telophase cell is). In supplemental figure 1 we show the data demonstrating these correlations. While this staging method is not perfect, it does allow us separate cells in early and late telophase. Quantifications shown in this figure are all done in late telophase cells as stages using these two criteria.

The splitting of the microtubules at the ends of the ICB is marked in the acetylated tub image, not the alpha tubulin; should not the microtubules in both show the splitting? This seems like it needs to be addressed. Could the apparent splitting be interphase microtubules in the daughter cells that overlap with the minus ends of ICB microtubules?

We see that quite often. The anti-acetylated tubulin antibody is a much better antibody, thus, thinner microtubule bundles (especially in fan-out regions) are typically much easier to detect with anti-acetylated tubulin antibody as compared to anti- α -tubulin antibody.

Most interestingly, in the KO cell in G, lower, there appear to be 2 cut sites, one on either side of the asterisk. Is this observed in the KO cells? Please comment.

These are not cuts. It is now well established in the field that in early/mid telophase cells can symmetrically compact tubulin on each side of the midbody. These sites (as was shown by EM in several previous papers from few labs) still have microtubules, but due to high compaction is not accessible to antibodies. We see that at equal frequency in both WT and KO cells.

5. in figure 8 data on abscission is shown with images for cut and not-cut control cells. Where are the images of the KO cells? Additionally, the control cells show very nice splitting of the microtubules of the ICB (example 2)! How frequent is this organization observed?

We have added examples of the KO cells with and without cut sites (Figure 7F). We initially did not include them as they look similar but just occur at a lower frequency, but we understand that is not clear for readers who did not see the images. For the splitting phenotype, we will clarify this in the manuscript. It is not that control cells never display any splitting or fanout prior to severing, but that the extent of it is much less extreme and trends towards a more uniform microtubule organization pattern.

Other

Page 5 top. Add a reference for no stability of ICB microtubules.

We have added references for microtubule stability as requested.

For mean intensities where was the background measured, especially in cases where non specific dots were observed.

We do measure background and background gets subtracted from the intensity values during calculation.

Supplemental figure 3. Why is there no alpha tubulin staining in the midzone (second image)? Why is the mask for the fourth image only a small circle in center - why not use all the midbody microtubule area?

It is usually hard to detect tubulin in the midzone of anaphase cells. Presumably it is due to the fact that microtubules there are very dynamic and, unlike kinetochore microtubules, not bundled. One can start detecting midzone (central spindle) microtubules during ingression (see new Figure 2) since they start forming bundles. That is the reason why we decided to only do quantification of microtubule glutamylation in ingressing cells rather than anaphase cells (Figure 2) in our revisions.

As suggested, we changed the mask to all of the ICB rather than just midbody in early telophase.

Supplemental figure 4. There is little or no midbody/ICB signal with the PolyE antibody. To examine role of CCP1 and 5, why not use rGT335 staining? Why would CCP 1 or 5 be expected to label nuclei? Presumably the YFP is small enough to enter nucleus non-specifically.

Fair point. We have replicated this experiment with rGT335 and used it in Supplemental Figure 4A-B.

As for the nuclear signal, polyglutamylation of nuclear proteins is well known and our nuclear measurements were part of that understanding that overexpression of CCPs may affect polyglutamylation in the nucleus. Regardless, it is not the focus of our manuscript, thus, we have taken it out.

Figure 2 B shows punctate staining with PolyE that looks non specific; Figure Suppl 7 shows some (dim or blurry) staining of ICB with Poly E. What is going on with the data - does this antibody reliably identify Poly E (glutamylation) on the MB or ICB microtubules? In Suppl Fig 7 why the bright intensity near nucleus (top row control cell) and the bright blob in KO cell second row?

We agree that PolyE staining in telophase cells is non-specific. However, there is clear PolyE signal associated with mitotic spindle in metaphase cells. We edited text to make that clear. We also removed Supplemental Figure 7 since these are the images of telophase cells and all signal is likely background.

Supplemental figure 8 (current Supplemental Figure 7), panel A, KO cell shows signal at MB; control does not. Please comment.

We and other laboratories have previously shown that ICB localized FIP3-endosomes are very dynamic, constantly entering and leaving the midbody. We observe this behavior in both control and TPGS1-KO cells. Thus, TPGS1 does not appear to have dramatic effect on FIP3-endosome trafficking during telophase.

Supplemental Figure 9 shows PRC staining (panel A) which is expected to localize at regions of antiparallel microtubule overlap. Surprisingly in the KO cells PRC is located distal to the zone of overlap in the region of the 'splay-out". Please comment. Also for the spastin, images are shown for control cells; please include KO cells. For the control cell bottom row multiple dots of spastin are observed; are there two cut sites? Were both dots of spastin quantified? Are multiple dots of spastin staining frequently observed?

PRC1 staining typically begins around the antiparallel overlap in early telophase and spreads through the whole ICB by late telophase. For spastin, yes often there are cut sites on both sides, one following the other later. For our quantification in Supplemental Figure 9, presence of any spastin on either or both sides were counted as a point for spastin localization, as long as abscission had not yet occurred. It is not unusual to see multiple dots post-abscission, however, pre abscission spastin is localized specifically to the cut site/s. We added more spastin images to reflect all these observations.

Dear Prof. Prekeris,

Thank you for the submission of your revised manuscript to our editorial offices. I have now received the reports from the two referees that I asked to re-evaluate the study, you will find below. As you will see, both referees now support publication of your study in EMBO reports. However, both have several comments and suggestions to improve the manuscript, I ask you to address in a final revised manuscript. Please also provide a final p-b-p-response to these referee points and the editorial requests below.

Editorial requests:

- In the author list on the title page the corr. author has superscripts 1 and 3. I think it should be 1 and 2. Please check.
 - We updated our journal's competing interests policy in January 2022 and request authors to consider both actual and perceived competing interests. Please review the policy <https://www.embopress.org/competing-interests> and add a statement regarding your competing interests to the manuscript. Please name this section 'Disclosure and Competing Interests Statement' and put it after the Acknowledgements section.
 - Please add up to five keywords to the manuscript and order the manuscript sections like this, using only these names: Title page - Abstract - Keywords - Introduction - Results - Discussion - Methods - Data availability section - Acknowledgements - Disclosure and Competing Interests Statement - References - Figure legends
 - Please remove the mention of a published dataset from the data availability section (DAS). This section is restricted to deposited datasets produced during the study. Please add this as data citations to the reference list and use appropriate callouts. See the section 'data citation' here:
<https://www.embopress.org/page/journal/14693178/authorguide#referencesformat>
 - Please use our reference format (et al needs to be used after 10 author names; DOIs should only be used for preprints and datasets that have not been published yet):
<http://www.embopress.org/page/journal/14693178/authorguide#referencesformat>
 - Please check again that the number "n" for how many independent experiments were performed, their nature (biological versus technical replicates), the bars and error bars (e.g. SEM, SD) and the test used to calculate p-values is indicated in the respective figure legends (main and Appendix figures). Please also check that all the p-values are explained in the legend, and that these fit to those shown in the figure. Please provide statistical testing where applicable. Please avoid the phrase 'independent experiment' but clearly state if these were biological or technical replicates. Please also indicate (e.g. with n.s.) if testing was performed, but the differences are not significant. In case n=2, please show the data as separate datapoints without error bars and statistics. See also:
<http://www.embopress.org/page/journal/14693178/authorguide#statisticalanalysis>
- If n<5, please show single datapoints for diagrams. It seems 'n.s.' is missing in all diagrams (also in the Appendix). Please add this. Moreover:
- Please note that the exact p values are not provided in the legends of figures 2C, 5C
 - Please note that the box plots need to be defined in terms of minima, maxima, centre, bounds of box and whiskers, and percentile in the legend of figure 3A
 - Please note that information related to n is missing in the legend of figure 3A.
- Please make sur all figure panels (main and Appendix figures) are called out separately and sequentially. Presently, Fig. 4A is called out before Fig. 3A and there seem to be no separate callouts for Fig. 5C and Fig. 7G. Please check.
 - It seems the primer information and the antibodies are part of the Reagents & Tools Table. There is no need to repeat this information in the Methods section. Please thus remove the primer table and the antibody information from the main manuscript text file.
 - Please add the title of the paper to the Appendix title page ('Appendix for ...') and use the nomenclature 'Appendix Figure Sx' for the Appendix items, for their name and for the callouts. Moreover, please make sure all Appendix Figures are called out in the main manuscript text and that the panels are called out separately and sequentially.
 - Please supply the Appendix file at higher resolution. Currently, the cells appear pixelated under analysis.
 - The table (Table 1) must not be provided as an image file - it needs to be editable, e.g. in Word or Excel.
 - Thanks for providing the source data. Please upload this as one folder per main figure, grouping together all the files for this figure (and ZIPed together), and one folder for the Appendix Figures, grouping together all the files for each Appendix Figure in

separate folders (and ZIPed together). Moreover, please provide the numerical data as Excel files.

In addition, I would need from you uploaded separately (please remove this from the manuscript text file):

Best,

Referee #1:

In their revised manuscript, Sachs et al have addressed most of my comments. There are a few edits that would strengthen the manuscript:

p.5: Monoclonal recombinant tubulin anti-glutamylated antibody rGT335 is known to detect tubulin glutamylation of 2 or more glutamates encompassing most glutamylated chains" is misleading. It recognizes the first glutamate added to the tubulin primary sequence regardless of how long the glutamylation chain is.

I still find figure 2 (was figures 2 and 3) confusing and both the figures and the text could do a better job walking the reader through the data and rationale.

Since the main point that the authors want to make is that the relative levels of long-chain vs short-chain glutamylation are changing as cells progress from metaphase to telophase, it is important to show the poly E data in the main figure.

In the last paragraph on p.5, the authors state "Our data suggest that progression from metaphase to telophase involves an increase in short-chain glutamylation" but the data in Figure 2B show that rGT335 is decreasing or, as the authors state in the previous paragraph "rGT335 signal remained constant in metaphase, early telophase, and late telophase microtubules." The authors should fix this discrepancy.

In the last paragraph on p.5, the authors want to test three possibilities but the immunostaining does not distinguish between these possibilities. Since the microtubules being stained in early and late telophase are not the same ones as in metaphase, the third possibility seems to be the obvious one. That is, the modifications on an individual microtubule do not change but rather the microtubules and their modification patterns change. The authors should rewrite the text to make this more clear.

An outstanding question is that TTL1 and TTL11 are thought carry out elongation of glutamate chains and should therefore lead to increased pole staining in the midbody. This should be discussed.

Many references are missing. For example, the statements about long-chain polyglutamylation regulating dynein and kinesin-based transport should cite Lessard et al 2019. The authors should go through the text to be sure they are citing all of the relevant literature.

Referee #2:

The authors have made significant change to their manuscript, improving the work in several ways. I have a few thoughts about the overall message and a few additional points that they could consider addressing.

Overall, the localization imaging for the TTL enzymes and TPGS1 shows very weak localization. One possibility is that the enzymes that they are localizing are present in very low concentration, or perhaps are only dynamically associated with the microtubules. Alternatively this is a fixation issue (fixed images) or as the authors mention, the cells response to overexpressed proteins (live imaging). It might be worth mentioning the low intensity of staining (or expression), because maybe it is telling us something about the quantity of protein needed for the task.

Staging of early and late telophase. I appreciate the challenges in staging fixed cells which the authors address in their response to reviewers. However, in Fig 3D both cells look to be at a similar stage, but one has TTLL1 GFP localized to midbody and the other (upper) does not. Both cells look rounded; microtubules are slightly longer in the lower cell. Could staging issues also affect results in the cells in panel C? One thought is to report the percentage of cells of a given morphology that show staining.

In figure 5A, KO cell with TTLL1-E326G, lower row; if the cell is KO for TPGS1, why no phenotype? Microtubules fanning out, as seen in the KO cell by live imaging in panel E and in figure 6?

The staining for PRC1 (control cell) in the supplemental figure appears to be inconsistent with the literature. PRC1 binds preferentially to antiparallel microtubules. This was shown for cells first in images from the Mitchison lab and later many other labs. The staining in the literature shows a band of PRC1 precisely where the microtubules overlap in the midzone and midbody. The control cell needs a panel of the microtubules so the reader can be sure the PRC1 is limited to the overlap, otherwise it is difficult to be sure. The PRC1 in the KO cell is all along the microtubules, which suggests that the microtubules are antiparallel, which is unexpected, but again the microtubule staining is not shown. This should be corrected prior to publication. The authors reference two PRC1 papers and both are in vitro studies - they need to cite a paper where PRC1 was localized in cells and show images with the expected distribution in control cells.

The authors also localize Spastin (suppl fig 9). some research shows pair of lines in the center of the midbody (one brighter the other dimmer; see PMC10908825). The staining shown in supplemental figure 9 is not as clear as shown by others and should be repeated. It is important to know if spastin is properly localized in the KO cells, because the phenotype is a defect in the organization of midzone microtubules and abscission

page 4. Central spindle microtubules (also called the interzone) arise from branched nucleation on kinetochore microtubules of each half spindle. This was shown in papers from the Tolic lab. Interzonal microtubules are not arising from the centrosome.

Figure 1 B. post-abscission image has asterisk - is this really the midbody? Looks like a blob. Do you have a clearer example?

Page 8 typo-fig E-G needs a figure number.

We thank the Referees for the constructive suggestions. We incorporated essentially all of them. Point-by-point changes are listed below. The text changes in manuscript are marked in yellow.

Referee #1:

1) p.5: Monoclonal recombinant tubulin anti-glutamylated antibody rGT335 is known to detect tubulin glutamylation of 2 or more glutamates encompassing most glutamylated chains" is misleading. It recognizes the first glutamate added to the tubulin primary sequence regardless of how long the glutamylation chain is.

Manuscript corrected and now states that "monoclonal recombinant tubulin anti-glutamylated antibody rGT335 is known to detect tubulin glutamylation of 1 or more glutamates encompassing most glutamylated chains".

2) I still find figure 2 (was figures 2 and 3) confusing and both the figures and the text could do a better job walking the reader through the data and rationale.

As suggested, we re-wrote Result section that discusses figure 2, as well as figure 2 legend.

3) Since the main point that the authors want to make is that the relative levels of long-chain vs short-chain glutamylation are changing as cells progress from metaphase to telophase, it is important to show the poly E data in the main figure.

As suggested, we moved polyE data to the main figure 2.

4) In the last paragraph on p.5, the authors state "Our data suggest that progression from metaphase to telophase involves an increase in short-chain glutamylation" but the data in Figure 2B show that rGT335 is decreasing or, as the authors state in the previous paragraph "rGT335 signal remained constant in metaphase, early telophase, and late telophase microtubules." The authors should fix this discrepancy.

We corrected the discrepancy and now text states that rGT335 signal remained constant in metaphase, early telophase, and late telophase microtubules.

5) In the last paragraph on p.5, the authors want to test three possibilities, but the immunostaining does not distinguish between these possibilities. Since the microtubules being stained in early and late telophase are not the same ones as in metaphase, the third possibility seems to be the obvious one. That is, the modifications on an individual microtubule do not change but rather the microtubules and their modification patterns change. The authors should rewrite the text to make this more clear.

We fully agree with the review that third possibility most likely, although our data does not fully prove it. We rewrote that section to make it more clear.

6) An outstanding question is that TTLL1 and TTLL11 are thought carry out elongation of glutamate chains and should therefore lead to increased pole staining in the midbody. This should be discussed.

As suggested we added a short discussion regarding that to the Discussion Section.

7) Many references are missing. For example, the statements about long-chain polyglutamylation regulating dynein and kinesin-based transport should cite Lessard et al 2019. The authors should go through the text to be sure they are citing all of the relevant literature.

We have added a citation to Lessard et al 2019. We also went through the manuscript to add more citation where they are needed.

Referee #2:

1) Overall, the localization imaging for the TTLL enzymes and TPGS1 shows very weak localization. One possibility is that the enzymes that they are localizing are present in very low concentration, or perhaps are only dynamically associated with the microtubules. Alternatively, this is a fixation issue (fixed images) or as the authors mention, the cells response to overexpressed proteins (live imaging). It might be worth mentioning the low intensity of staining (or expression), because maybe it is telling us something about the quantity of protein needed for the task.

Mention of low intensity (expression) of staining has been added.

2) Staging of early and late telophase. I appreciate the challenges in staging fixed cells which the authors address in their response to reviewers. However, in Fig 3D both cells look to be at a similar stage, but one has TTLL1 GFP localized to midbody and the other (upper) does not. Both cells look rounded; microtubules are slightly longer in the lower cell. Could staging issues also affect results in the cells in panel C? One thought is to report the percentage of cells of a given morphology that show staining.

As suggested, we analyzed the presence of TTLL1 in the MB in telophase cells showing different morphology. In 78.1% of flattened telophase cells TTLL1 was localized in the MB, while the same was only in 35.7% of round telophase cells. We added these data to the text of manuscript (Result chapter).

3) In figure 5A, KO cell with TTLL1-E326G, lower row; if the cell is KO for TPGS1, why no phenotype? Microtubules fanning out, as seen in the KO cell by live imaging in panel E and in figure 6?

The extend of the fanning out in fixed cells varies a bit. It is likely due to the fact that eventually even KO cells reorganize central spindle microtubules into anti-parallel bundles. Since these are fixed cells, we have no way of knowing how long it has been in late telophase until it got fixed.

4) The staining for PRC1 (control cell) in the supplemental figure appears to be inconsistent with the literature. PRC1 binds preferentially to antiparallel microtubules. This was shown for cells first in images from the Mitchison lab and later many other labs. The staining in the literature shows a band of PRC1 precisely where the microtubules overlap in the midzone and midbody. The control cell needs a panel of the microtubules so the reader can be sure the PRC1 is limited to the overlap, otherwise it is difficult to be sure. The PRC1 in the KO cell is all along the microtubules, which suggests that the microtubules are antiparallel, which is unexpected, but again the microtubule staining is not shown. This should be corrected prior to publication. The authors reference two PRC1 papers and both are in vitro studies - they need to cite a paper where PRC1 was localized in cells and show images with the expected distribution in control cells.

As suggested, we added panels of the microtubule staining to Figure S8.

5) The authors also localize Spastin (suppl fig 9). some research shows pair of lines in the center of the midbody (one brighter the other dimmer; see PMC10908825). The staining shown in supplemental figure 9 is not as clear as shown by others and should be repeated. It is important to know if spastin is properly localized in the KO cells, because the phenotype is a defect in the organization of midzone microtubules and abscission

We have repeated this staining but generally this is what we see. Only very occasionally do we see “pair of lines” staining. Perhaps that reflects the fact that these “pairs of lines” only form at certain stage of telophase. It is also possibly that staining looks different since we used different antibody and slightly different fixation method. In any case, we agree with the referee that we did not fully ruled out the possibility that KO may also affect spastin recruitment. We added a couple sentences to the Result section to point that out.

6) page 4. Central spindle microtubules (also called the interzone) arise from branched nucleation on kinetochore microtubules of each half spindle. This was shown in papers from the Tolic lab. Interzonal microtubules are not arising from the centrosome.

Our apologies for the mistake. The text referring to central spindle microtubules arising from centrosome has been removed from the manuscript.

7) Figure 1 B. post-abscission image has asterisk - is this really the midbody? Looks like a blob. Do you have a clearer example?

We have replaced with better example of post-abscission cell (see figure 1B).

8) Page 8 typo-fig E-G needs a figure number.

Figure number added.

Prof. Rytis Prekeris
University of Colorado Anschutz Medical Campus
Cell and Developmental Biology
12801 E. 17th Ave.
Aurora, CO 80045
United States

Dear Prof. Prekeris,

Thank you for the submission of your final revised manuscript. I now went through this and your final p-b-p-response and consider the remaining referee points and the editorial requests as adequately addressed.

I am thus very pleased to accept your manuscript for publication in the next available issue of EMBO reports. Thank you for your contribution to our journal.

Yours sincerely,
